# Macrophages in epididymal adipose tissue secrete osteopontin to regulate bone homeostasis

Bingyang Dai[1,6], Jiankun Xu[1,6], Xu Li[1,6], Le Huang[1], Chelsea Hopkins[1], Honglian Wang[2], Hao Yao[1], Jie Mi[1], Lizhen Zheng[1], Jiali Wang[3], Wenxue Tong [1], Dick Ho-kiu Chow [1], Ye Li [1], Xuan He[1], Peijie Hu[4], Ziyi Chen[1], Haiyue Zu[1], Yixuan Li[5], Yao Yao[5], Qing Jiang[5] & Ling Qin [1✉]

Epididymal white adipose tissue (eWAT) secretes an array of cytokines to regulate the metabolism of organs and tissues in high-fat diet (HFD)-induced obesity, but its effects on bone metabolism are not well understood. Here, we report that macrophages in eWAT are a main source of osteopontin, which selectively circulates to the bone marrow and promotes the degradation of the bone matrix by activating osteoclasts, as well as modulating bone marrow-derived macrophages (BMDMs) to engulf the lipid droplets released from adipocytes in the bone marrow of mice. However, the lactate accumulation induced by osteopontin regulation blocks both lipolysis and osteoclastogenesis in BMDMs by limiting the energy regeneration by ATP6V0d2 in lysosomes. Both surgical removal of eWAT and local injection of either clodronate liposomes (for depleting macrophages) or osteopontin-neutralizing antibody show comparable amelioration of HFD-induced bone loss in mice. These results provide an avenue for developing therapeutic strategies to mitigate obesity-related bone disorders.

[1] Musculoskeletal Research Laboratory of Department of Orthopaedics & Traumatology and Innovative Orthopaedic Biomaterial & Drug Translational Research Laboratory, Li Ka Shing Institute of Health Sciences, The Chinese University of Hong Kong, Hong Kong, China. [2] Research Center for Integrated Medicine, Affiliated Traditional Medicine Hospital of Southwest Medical University, 646000 Luzhou, Sichuan, China. [3] School of Biomedical Engineering, Sun Yat-sen University, Guangzhou, Guangdong, China. [4] Department of Biomedical Engineering, The Hong Kong Polytechnic University, Hung Hom, Hong Kong, China. [5] State Key Laboratory of Pharmaceutical Biotechnology, Department of Sports Medicine and Adult Reconstructive Surgery, Nanjing Drum Tower Hospital, The Affiliated Hospital of Nanjing University Medical School, Nanjing, Jiangsu, China. [6] These authors contributed equally: Bingyang Dai, Jiankun Xu, Xu Li. ✉email: lingqin@cuhk.edu.hk

Obesity has conventionally been considered to exert a favourable influence on bone mass, but recent clinical studies demonstrate that body weight itself negatively correlates with bone mass when excluding the confounding effect of gravity[1,2]. Growing epidemiological evidence offers an explanation that the metabolic syndrome of visceral obesity is responsible for bone loss through the secretion of pro-inflammatory cytokines and hormones, such as tumour necrosis factor alpha (TNF-α) and interleukin 1 beta (IL-1β)[3,4]. These released cytokines can metabolically regulate other organs and tissues, such as the liver, skeletal muscle, and cardiovascular system[4,5], while their regulations on bone metabolism is far from clear.

Although the role of white adipose tissue (WAT) in energy storage has been firmly established, we are only recently coming to appreciate the profound influence of visceral WAT (VAT) on metabolic homeostasis. The accumulation of ectopic adipose tissue surrounding the viscera is directly related to the development of insulin resistance and chronic inflammation[6–8]. Hypertrophy of adipocytes is associated with an increase in hypoxia experienced in VAT rather than subcutaneous WAT. Sometimes, hypoxic adipocytes undergo necrosis, leading to infiltration of immune cells and tissue inflammation[9–11]. These alterations may also lead to persistently elevated lipids in the blood and may contribute to earlier onset of metabolic disease with a 'recuperative' mass of epididymal WAT (eWAT)[9,12,13]. More concretely, the infiltrating adipose tissue-derived macrophages (ATMs) in VAT, especially eWAT, are a pathological feature of obesity, as these cells contribute to the inflammatory response by releasing various cytokines[4].

In a microarray analysis of eWAT taken from lean and obese mice, *spp1* (secreted phosphoprotein 1, or *osteopontin*, OPN) was the most highly upregulated gene (~28 fold) in mice fed with high-fat diet (HFD) for 12 weeks[14]. The increased OPN mediates ATM recruitment to eWAT[15]. OPN secreted predominantly by osteoblasts and osteocytes is an abundant non-collagenous phosphoprotein of the extracellular matrix of bone. It serves to regulate matrix mineralization and bone cell (osteocyte and osteoclast) attachment[16–19]. OPN is highly concentrated at cement lines where residual and newly formed bone integrate (mineral–matrix interactions) at bone surfaces in the lamina limitans structure surrounding osteocytes and underlying osteoclasts (cell–matrix interactions)[19]. OPN inhibits mineralization by stabilizing mineral precursor phases and binding to crystal surfaces[20], and OPN-deficient mice show increased mineral content and crystallinity[16,17]. OPN-dependent intracellular signalling is observed during initial osteoclast attachment to the bone surface and during the osteoclastic resorption phase[3]; thus, OPN plays an important role in the bone remodelling cycle. The systemic effects of OPN on bone homeostasis have not been well studied, especially from the aspect of eWAT biology. Here, we further investigated the links among bone, OPN, and eWAT.

Bone marrow adipose tissue (BMAT) can be divided into two subtypes: constitutive BMAT (cBMAT) and regulated BMAT (rBMAT). cBMAT is a contiguous group of adipocytes that predominate at distal skeletal sites, whereas rBMAT is interspersed with the haematopoietic bone marrow in the proximal tibia[21,22]. rBMAT adipocytes are more closely situated in areas of high bone turnover and are better positioned to actively influence bone remodelling[23]. Therefore, rBMAT is a dynamic compartment that performs lipolysis and lipogenesis related to ageing, inflammation, obesity, and type 2 diabetes, as well as being relevant in therapeutic contexts such as radiotherapy or glucocorticoid treatment[23].

Although all macrophages share similar immunological responses and phenotypes in inflammation, their functions are tissue-specific[24]. Bone-marrow-derived macrophages (BMDMs) differentiate into macrophages with ATM-like characteristics in the presence of adipose tissue in vitro, including neutral lipid accumulation, lysosome activation, and lipolysis[25,26]. In addition to autonomously maintaining ATMs in eWAT, OPN mediates ATM infiltration in HFD-fed mice by promoting the recruitment of macrophages, some of them BMDMs[15,24,27]. Therefore, BMDMs also have a similar function of recycling the lipids released from rBMAT within the bone marrow, and this process is regulated by OPN. Lysosomal-related V-ATPase subunit ATP6V0d2 is a key macrophage-specific component of cytoskeletal rearrangements for osteoclastogenesis and lipid metabolic activation[28–30]. We hypothesized that eWAT-secreted OPN accumulated in the bone marrow to mediate bone resorption and lipid recycling via lysosomal-related ATP6V0d2 activity in BMDMs.

Here, we show that eWAT-secreted OPN induces lipophagocytic mobilization of BMDMs to a lipid-rich pool in the bone marrow, while OPN also promotes bone matrix degradation by activating osteoclasts in HFD-fed mice, whereas the lactate accumulation resulting from OPN regulation blocks both lipolysis and osteoclastogenesis in BMDMs by limiting the energy regeneration by ATP6V0d2 in lysosomes. Surgical removal of eWAT, local injection of clodronate liposomes (to deplete macrophages), and local injection of OPN-neutralizing antibody show comparable effects on ameliorating HFD-induced bone loss in mice. These results provide an avenue for developing therapeutic strategies to modulate VAT inflammation and thus ameliorate obesity-related bone loss.

## Results

**eWAT alters the trabecular bone and rBMAT mass in HFD-fed mice.** The body weight began to diverge at week 2 when the HFD-fed group showed a significantly greater body weight than the normal-fat diet (NFD)-fed group (Fig. 1a). eWAT showed a significantly greater mass at week 4 in the HFD-fed group than in the NFD-fed group, and its mass peaked at week 12 (Fig. 1b). Inguinal white adipose tissue (iWAT) showed rapid accumulation in the HFD-fed group without tapering off (Fig. 1b). micro-computed tomography (μCT) analysis demonstrated a significantly greater mass of rBMAT in the HFD-fed group than in the NFD-fed group at week 8, and it peaked at week 12 (Fig. 1c, Supplementary Fig. 1a). μCT analysis of the trabecular bone at the proximal tibia showed that the bone volume fraction (i.e. bone volume (BV) over total volume (TV), BV/TV) and bone mineral density (BMD) were significantly lower in the HFD-fed group than in the NFD-fed group starting at weeks 8 and 4, respectively (Fig. 1d, Supplementary Fig. 1b, c). Linear regression analysis showed that body weight was negatively correlated with BV/TV (Fig. 1e, Supplementary Fig. 1c). The significant differences in eWAT and iWAT mass between the NFD and HFD-fed groups arose at week 4, while the alterations in rBMAT and BV/TV were only observed starting at week 8. At week 8, insulin tolerance tests (ITTs) showed a weaker response in the HFD-fed group than the NFD-fed group (Fig. 1f). These clues suggest that both eWAT and iWAT might induce bone loss while rBMAT accumulated, independent of the insulin tolerance.

The increased presence of crown-like structures (CLSs) in the stromal vascular fraction of adipose tissues is an important predictor of chronic inflammation and is associated with the clearance of necrotic adipocytes[25,31]. Notably, the percentage of CLS area in the eWAT of the HFD-fed group was significantly greater than that of the NFD-fed group at week 8, peaking at week 12 (Fig. 2a). Of note, the difference between the two groups abated at week 16 (Fig. 2a). The percentage of CLS area in iWAT

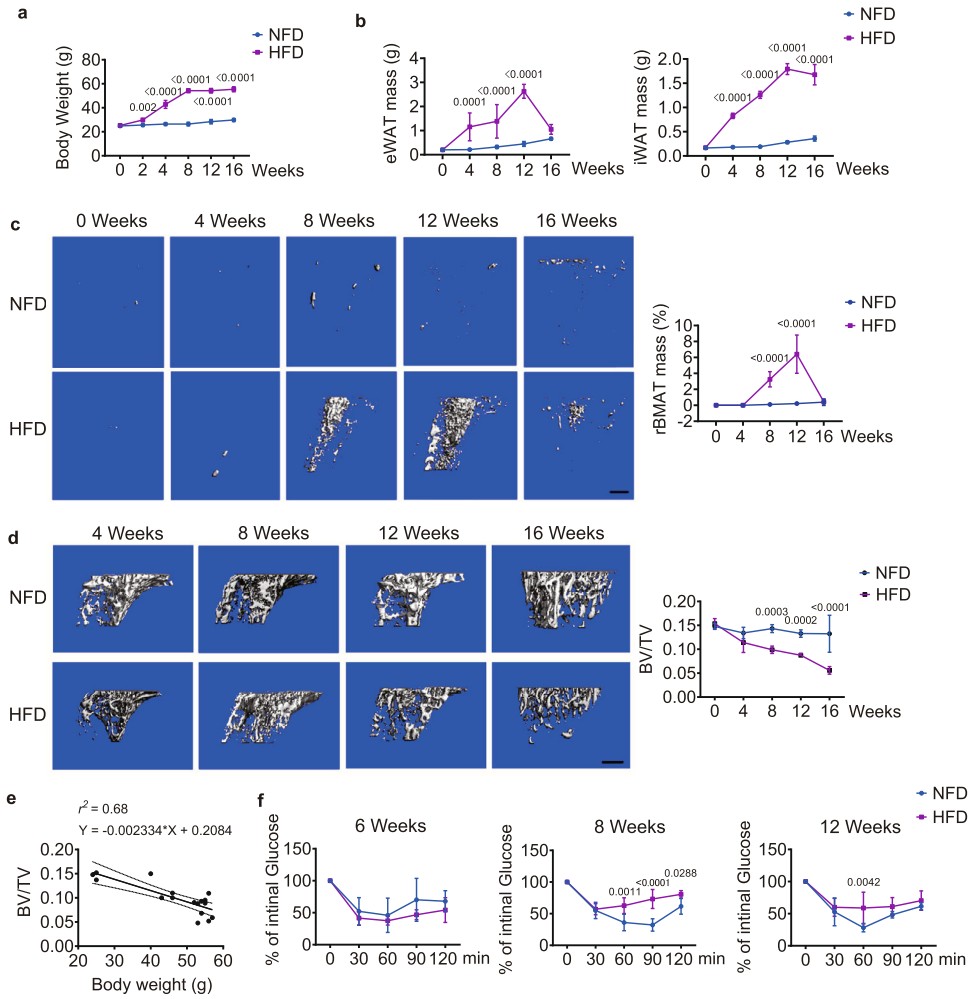

**Fig. 1 A high-fat diet induces alterations of adipose tissue and trabecular bone. a** Comparison of the body weight between NFD- and HFD-fed mice (*n* = 5 biologically independent samples). **b** Quantification of the mass of eWAT (*n* = 5 biologically independent samples) and iWAT (*n* = 5 biologically independent samples) of the NFD- and HFD-fed groups. **c** Representative μCT images (left) and quantification (right, *n* = 4 biologically independent samples) of rBMAT in the proximal tibiae of the NFD- and HFD-fed groups at the indicated time points. Scale bar: 500 μm. **d** Representative μCT images (left) and quantification of bone mass in proximal tibiae (right, *n* = 5 biologically independent samples). Scale bar: 500 μm. **e** Linear regression analysis of the BV/TV and body weight in HFD-fed mice (*n* = 17 biologically independent samples). **f** ITTs were performed at weeks 6, 8, and 12 in the NFD- and HFD-fed groups (*n* = 5 biologically independent samples). Images are representative of 3 independent experiments. All data are presented as mean ± SD. Two-way ANOVA with Sidak's post-hoc test (**a**–**d** and **f**) were used. Source data are provided as a Source Data file.

was similar between the two groups at all time points, indicating a similar inflammatory response in iWAT of both NFD- and HFD-fed mice (Fig. 2b). Consistent with the greater CLS area, the analysis of mRNA expression demonstrated that eWAT in the HFD group had significantly greater inflammatory gene expression (*Tnfa*, *Il-1b*, and *Il-10*), especially at week 12, than that in the NFD-fed group (Fig. 2c). No such differences were observed in the iWAT or bone marrow between the two groups, except for *Il-1b* and *Il-10*, which were slightly higher in the bone marrow of the NFD-fed group than the HFD-fed group at week 4 (Fig. 2d, e). The quantification of CLS area and inflammatory gene expression in other VAT depots, including perirenal white adipose tissue (PAT) and mesenteric white adipose tissue (MAT), were also absent, echoing the homeostatic disorders in mice fed a HFD (Supplementary Fig. 2a–d). These findings suggest that inflammation in eWAT might induce metabolic disorders of bone marrow, but not other AT depots.

When we surgically removed the bilateral eWAT depots (eWATx) of 12-week-old mice before starting the indicated diet (NFD or HFD), the trabecular bone mass in the HFD + eWATx

group was significantly greater than that of the HFD + Sham group at week 12 post-surgery instead of at week 8 (Fig. 3a, b). The rBMAT mass in the HFD + eWATx group was also significantly greater than that in the HFD + Sham group at week 8 (Fig. 3c, d). However, surgical removal of bilateral PAT depots (PATx) from 12-week-old mice before starting the indicated diet did not change the bone mass (Supplementary Fig. 3a, b). These findings indicated an intrinsic relationship between eWAT and bone marrow metabolism under HFD feeding conditions. In addition, this procedure of eWATx failed to normalize insulin sensitivity in the HFD-fed group, again suggesting that insulin tolerance was not the major factor in the bone marrow disorders induced by eWAT in HFD-fed mice (Fig. 3e).

**F4/80+ and CD11b+ macrophages in eWAT secrete OPN.**
Obesity induces macrophages to infiltrate the CLSs of eWAT, and the presence of these ATMs is closely correlated with lipid turnover and chronic systemic inflammation[25]. It was unknown whether inhibition of these actions by depleting ATMs would be able to restore bone marrow homeostasis. After injection of

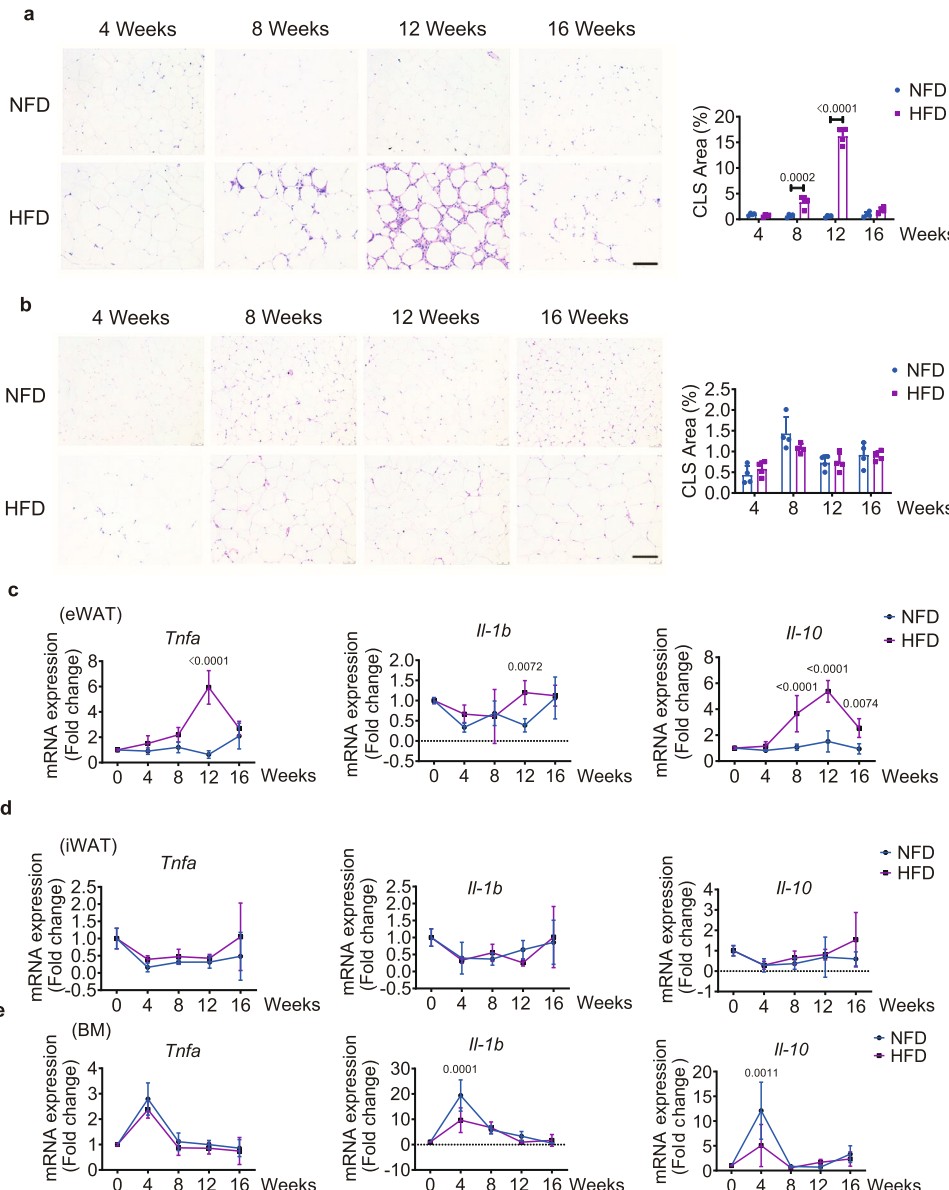

**Fig. 2 A high-fat diet induces inflammation in eWAT. a** Representative H&E staining (left) and quantification of CLS area percentage (right, $n = 4$) of eWAT from NFD- and HFD-fed mice at the indicated time points. Scale bar: 100 μm. **b** Representative H&E staining (left) and quantification of CLS area percentage (right, $n = 4$) of iWAT from NFD- or HFD-fed mice at the indicated time points. Scale bar, 100 μm. **c–e** Relative expression of *Tnfa*, *Il-1b*, and *Il-10* in eWAT ($n = 4$) (**c**), iWAT ($n = 4$) (**d**), and bone marrow (BM, $n = 4$) (**e**) over the course of the feeding regimen. $n = 4$ biologically independent samples per group. Images are representative of 3 independent experiments. All data are presented as mean ± SD. Two-way ANOVA with Sidak's post-hoc test (**a–e**) were used. Source data are provided as a Source Data file.

clodronate liposomes (CL, 0.675 mg/unilateral/3 days, to clear the ATMs) into the bilateral eWAT for 8 weeks, infiltration of ATMs was significantly lower in the HFD-fed mice (HFD + CL) than the HFD-fed mice treated with control liposomes (HFD + Ctrl) (Fig. 4a, b). The progression of bone loss was attenuated, and rBMAT mass significantly decreased in the HFD + CL group compared with the HFD + Ctrl group (Fig. 4c–e). Together, these results suggest that obesity activates the ATMs of eWAT to influence bone marrow metabolism.

To gain insight into the mechanism of the metabolic and inflammatory effects of eWAT on bone homeostasis, we analysed the mRNA expression in the whole eWAT of NFD- and HFD-fed mice at week 12, including inflammatory, lysosomal, and lipogenic markers. *Spp1* (osteopontin) was the most highly upregulated gene in eWAT of HFD-fed mice (Fig. 4f), consistent

with the microarray data from the Gene Expression Omnibus (accession number: GSE27017). OPN belongs to a family of secreted acidic proteins whose actions modulate mineralization in bone[32]. OPN plays a key role in linking obesity to the development of chronic inflammation by inducing the accumulation of macrophages in eWAT[15]. We found that the expression of *Spp1* in eWAT significantly increased with prolonged HFD feeding compared with NFD feeding (Fig. 4g). However, the expression of *Spp1* was similar between the NFD- and HFD-fed groups in the whole iWAT as well as PAT and MAT, except for the PAT collected at week 16, where *Spp1* was significantly higher in the HFD-fed group (Fig. 4g, Supplementary Fig. 4a, b).

Almost all ATMs in eWAT express F4/80 and CD11b and can secrete OPN[15,25]. We found that OPN was predominantly scattered around the F4/80+ macrophages in the stromal vascular

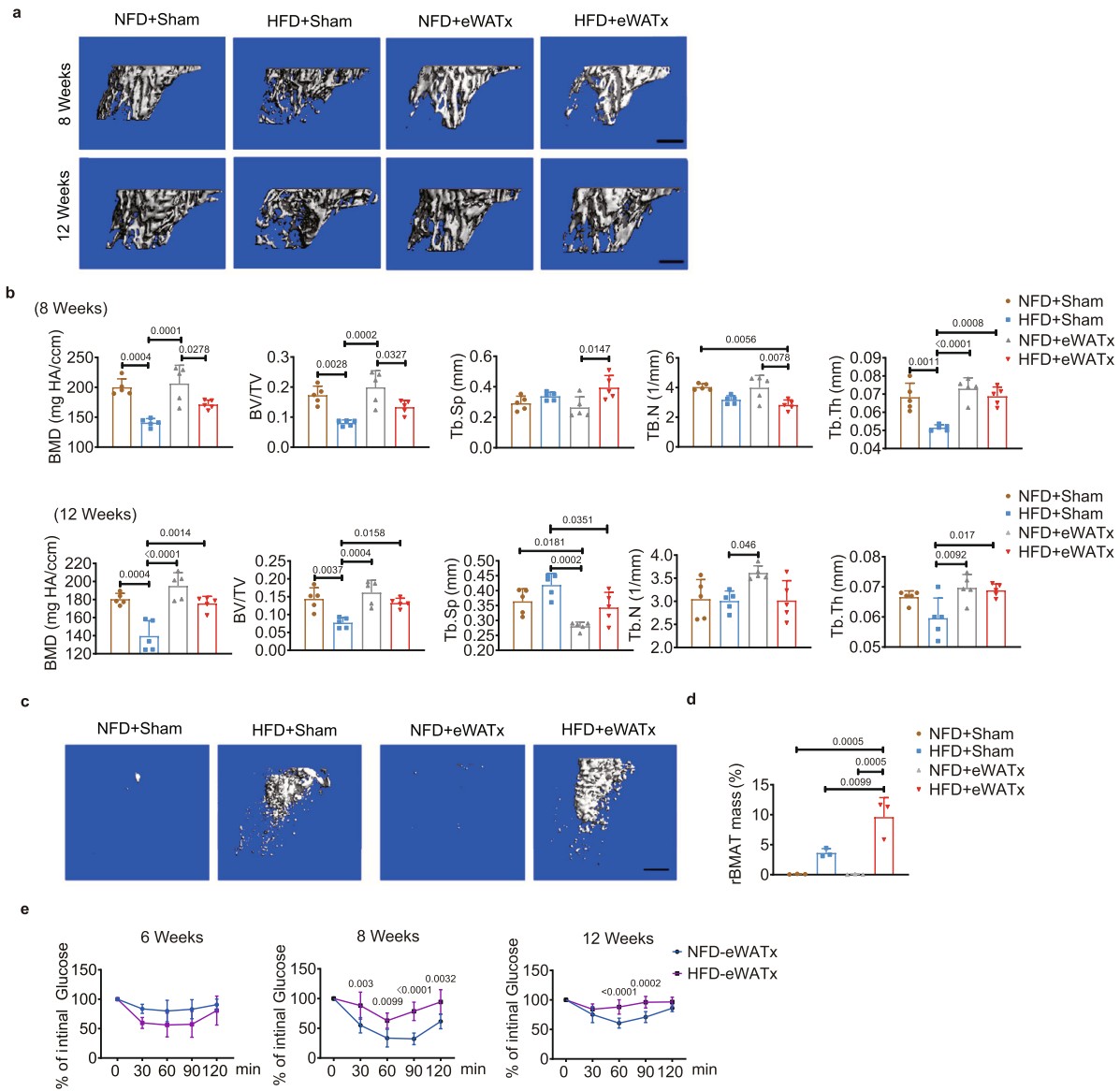

**Fig. 3 Removal of bilateral eWAT (eWATx) alters bone mass and rBMAT mass. a, b** Representative μCT images (**a**) and quantification (*n* = 5 biologically independent samples) (**b**) of proximal tibiae from NFD- and HFD-fed mice that had undergone either sham surgery or removal of the bilateral eWAT (eWATx). Scale bar: 500 μm. **c, d** Representative μCT images (**c**) and quantification (*n* = 3 biologically independent samples) (**d**) of rBMAT in tibiae at week 8 of feeding and eWATx or not. Scale bar: 500 μm. **e** ITTs were performed at weeks 6, 8, and 12 of NFD- and HFD-fed mice that had undergone eWATx (*n* = 5 biologically independent samples). Images are representative of 3 independent experiments. All data are presented as mean ± SD. Two-way ANOVA with Tukey's post-hoc test (**b, d**) and two-way ANOVA with Sidak's post-hoc test (**e**) were used. Source data are provided as a Source Data file. See also Supplementary Tables 2 and 3.

fraction (SVF), and it continuously spread up to week 12 of feeding (Fig. 5a, b, and Supplementary Fig. 5a). We isolated the SVF and adipocyte fraction (AF) from adipose tissues of HFD-fed mice at week 12 and found that the expression of *Spp1* in the SVF of eWAT was significantly higher than that in the AF (Fig. 5c). The isolated SVF was digested, and cell sorting was conducted by binning macrophages into the F4/80⁺CD11b⁺ ([FC+]) ATM group and the double-negative ([FC−]) ATM group (Fig. 5d and Supplementary Fig. 5b). The proportion of FC+ macrophages increased approximately four-fold in the HFD-fed group compared with the NFD-fed group at week 12 (Fig. 5e). *Spp1* mRNA was significantly upregulated in the FC+ subgroup over the FC− subgroup at week 12 in the HFD-fed group (Fig. 5f). Together, these results suggest that OPN is mainly derived from FC+ ATMs in eWAT, especially under HFD feeding conditions.

**eWAT-secreted OPN accumulates in bone marrow**. To verify this possibility of eWAT-secreted OPN executing function in bone marrow, we detected the proportion of this endogenous OPN in the bone marrow. First, there was no difference in OPN levels between the two groups, as determined by immunohisto-chemical staining of histological samples of tibiae and enzyme-linked immunosorbent assay (ELISA) of bone marrow super-natant (Fig. 6a, b and Supplementary Fig. 6). Second, although *Spp1* mRNA was similarly expressed in both FC+ BMDMs with and without HFD feeding, the proportion of FC+ BMDMs was significantly lower in the HFD-fed group than in the NFD-fed group at week 12 (Supplementary Fig. 7a–c). We also observed significantly less *Spp1* mRNA expression in the bone marrow of HFD-fed mice than NFD-fed mice (Fig. 6c), implying that cir-culating OPN might be an endogenous supplement for the

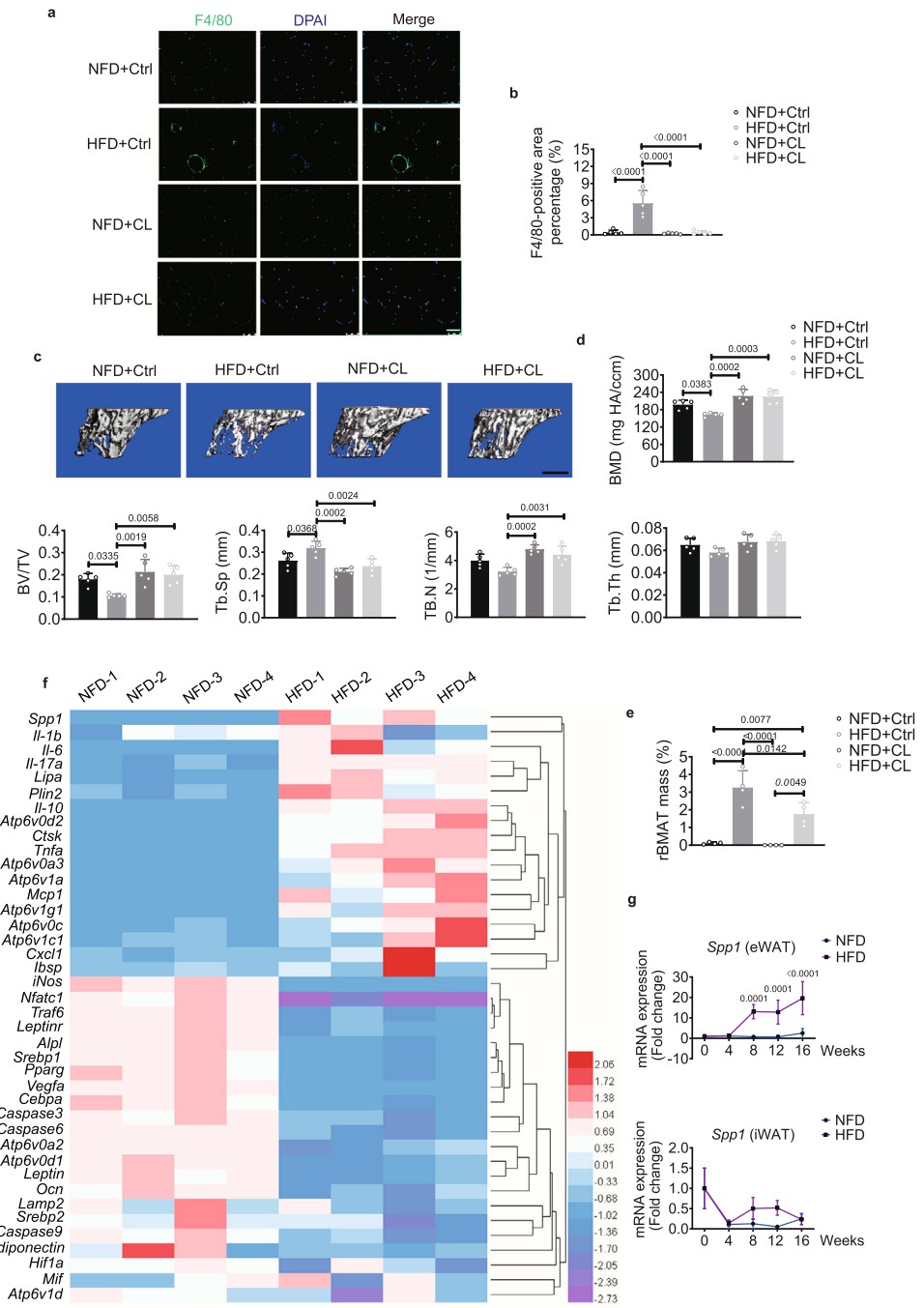

**Fig. 4 Infiltrating ATMs in eWAT are related to bone marrow metabolism. a** Immunofluorescence staining of macrophages (F4/80) in eWAT after mice were injected with clodronate liposomes (CL) or control liposomes (Ctrl) into bilateral eWAT for 8 weeks. Green, F4/80; blue, DAPI stain for cell nuclei. Scale bar: 75 µm. **b** Quantification of the F4/80-positive area percentage with immunofluorescence staining from A (*n* = 5 biologically independent samples). **c**, **d** Representative µCT image (**c**) and quantification (*n* = 5 biologically independent samples) (**d**) of proximal tibiae after mice were injected with control liposomes or CL for 8 weeks. Scale bar: 500 µm. **e** µCT quantification of rBMAT in tibiae after mice were injected with control liposomes or CL for 8 weeks (*n* = 4 biologically independent samples). **f** Heatmap summarizing the fold changes in mRNA expression of different biomarkers in eWAT from NFD- and HFD-fed mice at week 12 (*n* = 4 biologically independent samples). **g** Relative expression of *Spp1* in eWAT (*n* = 4 biologically independent samples) and iWAT (*n* = 4 biologically independent samples) over the course of feeding. Images are representative of 3 independent experiments. All data are presented as mean ± SD. Two-way ANOVA with Tukey's post-hoc test (**b**, **d**, **e**) and two-way ANOVA with Sidak's post-hoc test (**g**) were used. Source data are provided as a Source Data file. See also Supplementary Tables 4 and 5.

scarcity of bone marrow. Although the OPN concentrations in serum between the NFD- and HFD-fed groups were similar at different time points (Supplementary Fig. 8), we found that the concentration of OPN in serum, as well as in bone marrow supernatant, of the HFD + eWATx group was significantly lower

than that in the HFD + Sham group at week 12 (Fig. 6d, e), while the PATx group had a comparable concentration of OPN in serum to the Sham group with or without HFD treatment (Supplementary Fig. 9a, b). The fall in OPN concentrations in serum and bone marrow supernatant after eWATx indicated that

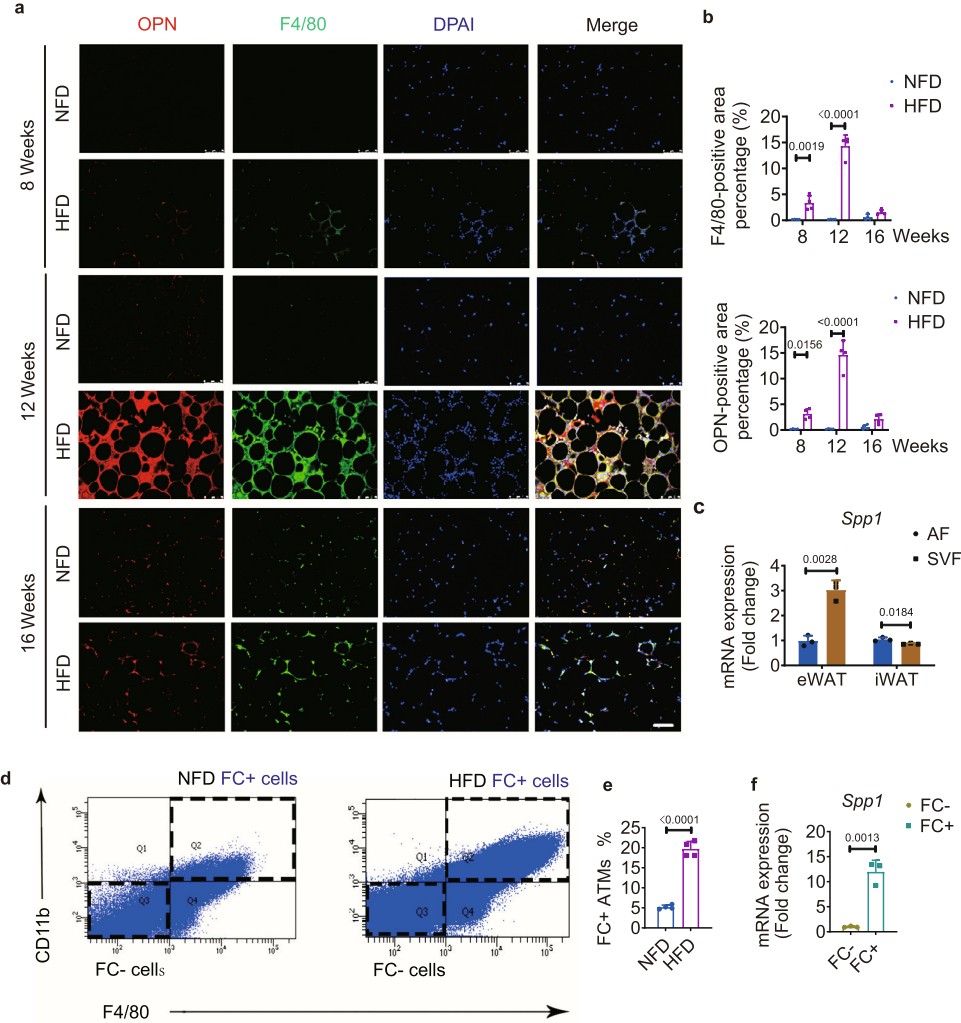

**Fig. 5 OPN is primarily secreted by F4/80 + and CD11b+ macrophages in eWAT. a** Representative images of immunofluorescence staining of OPN in eWAT from NFD- and HFD-fed mice at weeks 8, 12, and 16. Scale bar: 100 μm. **b** Quantification of the F4/80-positive or OPN-positive stained area percentage in eWAT from **a** (n = 4 biologically independent samples). **c** Relative expression of *Spp1* in the adipocyte fraction (AF) and stromal vascular fraction (SVF) from eWAT or iWAT of HFD-fed mice at week 12 (n = 3 biologically independent samples). **d** ATMs gated by FACS for F4/80 and CD11b expression from eWAT at week 12 of NFD (left) or HFD feeding (right). Q2 represents the F4/80 + CD11b + (FC + ) ATM population, and Q3 represents the F4/80−CD11b− (FC−) ATM population. **e** Percentage of FC + ATMs from NFD- and HFD-fed mice at week 12 (n = 4 biologically independent samples). **f** Relative expression of *Spp1* in FC+ and FC− ATMs from HFD-fed mice at week 12 (n = 3 biologically independent samples). Images are representative of 3 independent experiments. All data are presented as mean ± SD. Two-way ANOVA with Sidak's post-hoc test (**b**) and two-tailed Welch's *t*-test (**c**, **e**, **f**) were used. Source data are provided as a Source Data file.

eWAT might be a major source of OPN for the circulation and for bone marrow at this time point.

Further, we found that the concentration of OPN in the eWAT-derived conditioned medium from HFD-fed mice was significantly higher than that from NFD-fed mice at all time points (Fig. 6f), whereas iWAT-derived conditioned medium had a similar OPN concentration between the two groups (Fig. 6g). We injected recombinant human OPN (rOPN) conjugated with FITC into the left-side eWAT of HFD-fed mice, and 1 h later, the FITC signal was predominantly localized at both hind limbs and eWAT, but no other adipose tissues (Fig. 6h). At 24 h post-injection, the fluorescence was detectable in the spine, bilateral hind limbs, and eWAT (Fig. 6h). When we performed immunohistochemical staining using a specific anti-human OPN antibody, we found that staining of exogenous rOPN was scattered in the bone marrow environment of the vertebrae, tibiae, and femora at 1 h post-injection (Fig. 6i). At 24 h, this exogenous pool of rOPN was mainly accumulated on the surface

of trabecular bone (Fig. 6i). Even though we injected rOPN into the left-side eWAT, we found that rOPN was detectable in the bilateral eWAT at both time points (Fig. 6i). Examination of visceral organs and iWAT did not show positive rOPN staining (Fig. 6j). Endogenous OPN secreted by the mice was negative in these organs and tissues (Supplementary Fig. 10a, b). These results suggest that OPN in eWAT might selectively accumulate in the bone marrow through the circulatory system to compensate for OPN synthesis in the bone marrow.

**OPN participates in bone resorption.** Bone marrow mesenchymal stem cells (BMSCs), BMDMs, and mature osteoclasts are all involved in bone remodelling. We measured the expression of integrin αvβ3, the OPN receptor distributed in these cells. We found the highest expression of integrin αvβ3 in osteoclasts (Fig. 7a), suggesting that osteoclasts might also be a main target cell of OPN. Immunohistochemical staining for OPN and cathepsin K (CTSK; a lysosomal protease predominantly

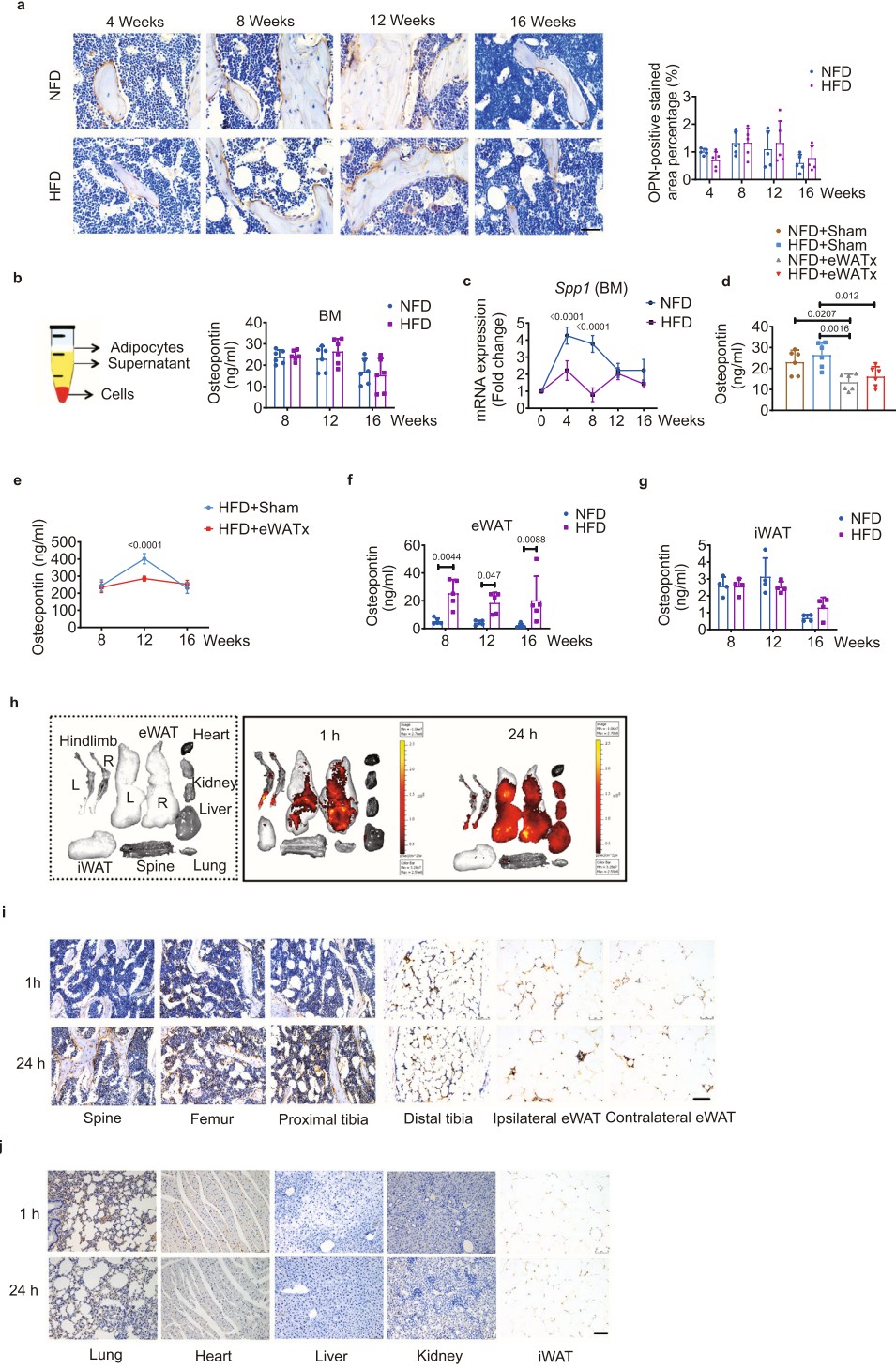

expressed in osteoclasts) in serial sections clearly demonstrated that OPN bound to osteoclasts on the surface of trabecular bone (Fig. 7b). rOPN improved the ability of osteoclasts to degrade bone matrix, as shown by an amplified resorption area and increased *Mmp9* expression in vitro (Fig. 7c–f). We also found that the activation of MMP9, as well as of integrin αvβ3, in the bone marrow of the HFD-fed group at week 12 was significantly higher than that of the NFD-fed group (Fig. 7g). The positive IHC staining intensity of MMP9 was lower in the bone marrow of HFD-fed mice treated with OPN-neutralizing antibody (Neu Ab) than in that of HFD-fed mice treated with saline (Fig. 7h). Bone loss was attenuated after injection of Neu Ab, accompanied by

fewer ATMs infiltrating eWAT and less accumulation of rBMAT (Fig. 7i–l and Supplementary Fig. 11a). The decreased Trap+ osteoclast surface area in the proximal tibiae also contributed to the greater bone mass of HFD-fed mice treated with Neu Ab (Supplementary Fig. 11b, c). Neu Ab did not increase bone mass in the mice fed the NFD, indicating that the benefits of OPN Neu Ab depended on the insult provided by the HFD (Fig. 7i, j). We injected Neu Ab conjugated with FITC into the left-side eWAT of HFD-fed mice. At 1 h or 24 h post-injection, the FITC signal was all near the injection site rather than in other AT depots and organs (Supplementary Fig. 11d), indicating that the Neu Ab injected into unilateral eWAT did not spread in situ. These results

**Fig. 6 eWAT-secreted OPN circulates to the bone marrow. a** Representative images of immunohistochemical staining of OPN (left) and the OPN-positive stained areas (right, $n = 5$ biologically independent samples) of the proximal tibiae of NFD- and HFD-fed groups. Scale bar: 50 μm. **b** Schematic representation of the isolation of bone marrow supernatant for ELISA (left). Free OPN levels in the bone marrow supernatant of NFD- and HFD-fed mice (right, $n = 6$ biologically independent samples). **c** Relative expression of *Spp1* in bone marrow over the course of feeding ($n = 4$ biologically independent samples). **d** OPN expression in bone marrow supernatant from mice with or without eWATx at week 12 of the indicated diet ($n = 6$ biologically independent samples). **e** OPN expression in the serum of the HFD-fed subgroups with and without eWATx at weeks 8, 12, and 16 ($n = 5$ biologically independent samples). **f** OPN concentration in eWAT (from NFD- and HFD-fed mice at weeks 8, 12, and 16)-derived conditioned medium ($n = 5$ biologically independent samples). **g** OPN concentration in iWAT (from NFD- and HFD-fed mice at weeks 8, 12, and 16)-derived conditioned medium ($n = 4$ biologically independent samples). **h** Ex vivo imaging (IVIS200 system) of the indicated organs and tissues collected 1 h (left part of the right box) or 24 h (right part of the right box) after unilateral eWAT injection of rOPN-FITC derived from HFD-fed mice at week 12. **i, j** Representative images of immunohistochemical staining of OPN (anti-human OPN antibody) in the spine, femur, tibia (proximal and distal), eWAT (ipsilaterally injected depot and contralateral depot) (**i**), lung, heart, liver, kidney and iWAT (**j**) of mice injected with rOPN-FITC into the unilateral eWAT for 1 h (top row) and 24 h (bottom row), respectively. Scale bar: 100 μm. BM: bone marrow. Images are representative of 3 independent experiments. All data are presented as mean ± SD. Two-way ANOVA with Sidak's post-hoc test (**a–c**, **e–g**) and two-way ANOVA with Tukey's post-hoc test (**d**) were used. Source data are provided as a Source Data file. See also Supplementary Table 6.

suggest a pathway by which eWAT-secreted OPN enhances bone loss through the promotion of osteoclast-mediated bone matrix degradation.

**OPN regulates BMDMs through ATP6V0d2.** We also found a higher expression of αvβ3 in BMDMs than in BMSCs (Fig. 7a), suggesting that BMDMs might be another main target of OPN. The macrophage-specific isoform of ATP6V0d2, a member of the ATPase family, plays a critical role in the maturation of osteoclasts[28]. We found that the expression of *Atp6v0d2* was significantly inhibited in the presence of rOPN, while other subunits were not affected (Fig. 8a). The protein level of ATP6V0d2 was also significantly lower in osteoclasts treated with rOPN in vitro (Fig. 8b). ATP6V0d2 is an ATP-dependent multimeric enzyme that pumps protons from the cytosol into the lumen of intracellular organelles, thereby controlling the acidification of lysosomes, endosomes, trans-Golgi network, and other intracellular vesicles[33]. In BMDMs, the liberation of ATP during osteoclastogenesis was significantly lower with the addition of rOPN (Fig. 8c). Lactate is a well-known product of glycolysis, and we found that its levels were significantly higher in osteoclasts treated with rOPN (Fig. 8d). We next blocked ATP consumption with antagonist 27 (an antagonist of integrin αvβ3) and found a significant increase in ATP accumulation and impaired accumulation of lactate in these BMDMs (Fig. 8c, d). An increase in glycolytic products suppresses the energy-regenerating capacity and the reactions downstream of this metabolic pathway[34], suggesting that the accumulated lactate in the present study might restrain the energy consumption of cytoplasmic metabolism via ATPase-dependent machinery. We incubated BMDMs with a gradient concentration of lactate during osteoclastogenic induction and found that lactate significantly inhibited *Atp6v0d2* expression in a dose-dependent manner, without any effect on other subunits (Fig. 8e). The inhibition of osteoclastogenesis was rescued by siRNA *Cd61* when BMDMs were treated with rOPN (Fig. 8f, g and Supplementary Fig. 12). These results support the notion that OPN inhibits the process of osteoclastogenesis by inducing the accumulation of lactate, which might prevent the regeneration of energy by ATP6V0d2.

**OPN induces lysosomal-dependent lipid metabolism in BMDMs.** Although the OPN level in bone marrow was similar in the NFD- and HFD-fed groups (Fig. 6b), the expression of integrin αvβ3 in the bone marrow of the HFD-fed group was significantly higher than that of the NFD-fed group at week 12 (Fig. 7g). It is unclear whether the increased expression of

integrin αvβ3 was related to OPN-dependent lipid metabolism within the bone marrow. We found that the expression of integrin αvβ3 was elevated in metabolically activated macrophages (MMes), supporting the notion that OPN could enhance macrophage migration to the lipid pool in vitro (Fig. 9a–d). Given that activated ATMs are involved in phagocytosing excess lipids in eWAT, we reasoned that BMDMs would also phagocytose lipids and float in the bone marrow supernatant because they would contain a large amount of lipid. Immunofluorescence staining showed lipid-laden F4/80+ macrophages were present in buoyant samples from the bone marrow supernatant of the HFD-fed group, as indicated by the expression of perilipin 1, a marker of lipid droplets (Fig. 9e). We next sorted the BMDMs by binning FC+ macrophages into the perilipin 1+ group and perilipin 1− group (Fig. 9f and Supplementary Fig. 13). The proportion of FC+ and perilipin 1+ BMDMs was statistically greater in the HFD-fed group than in the NFD-fed group at weeks 8 and 12 (Fig. 9g), and this relationship between the two groups was similar to that of free fatty acid (FFA) in bone marrow (Supplementary Fig. 14). Together, these results suggest that a high concentration of lipids in the bone marrow activates integrin αvβ3 expression and that OPN accelerates lipophagocytosis of BMDMs.

The lactate concentration in bone marrow was statistically greater in the HFD-fed group than in the NFD-fed group at weeks 8 and 12 (Fig. 9h). We also found that the lipid content in BMDMs incubated with Neu Ab and eWAT (bottom) was significantly lower than that without Neu Ab treatment (middle) in vitro (Fig. 9i, j). eWAT-secreted OPN in this co-culture system also induced significantly lower ATP and higher lactate than the group treated with Neu Ab (Fig. 9k). These results suggest that OPN-regulated lipid activation in the bone marrow of HFD-fed mice results in a higher concentration of lactate than that in the NFD-fed group.

ATP6V0d2 has two functions. It plays a critical role in the maturation of pre-osteoclasts, and it is also indispensable for lysosomal-dependent lipolysis in macrophages[25]. We found that the expression of *Atp6v0d2* was significantly elevated in MMes but significantly inhibited in the presence of rOPN (Fig. 9l). Obesity activates lysosome biogenesis, which is involved in lipolysis in eWAT, such as in the expression of the vacuolar (H+)-ATPase ATP6V0d2, the structural lysosome protein lysosome-associated membrane protein 2 (LAMP2), and the acid lipase called lipase A (LIPA)[25]. Knockdown of *Cd61* (which encodes the integrin subunit β3) in MMes using siRNA rescued the inhibitory effect of rOPN on the expression of *Atp6v0d2*, *lamp2*, and *lipa*, indicating that OPN arrested lipolysis in lipid-laden BMDMs (Supplementary Fig. 15a–c). These results together indicate that OPN induced lipophagocytosis in BMDMs, which was accompanied by robust

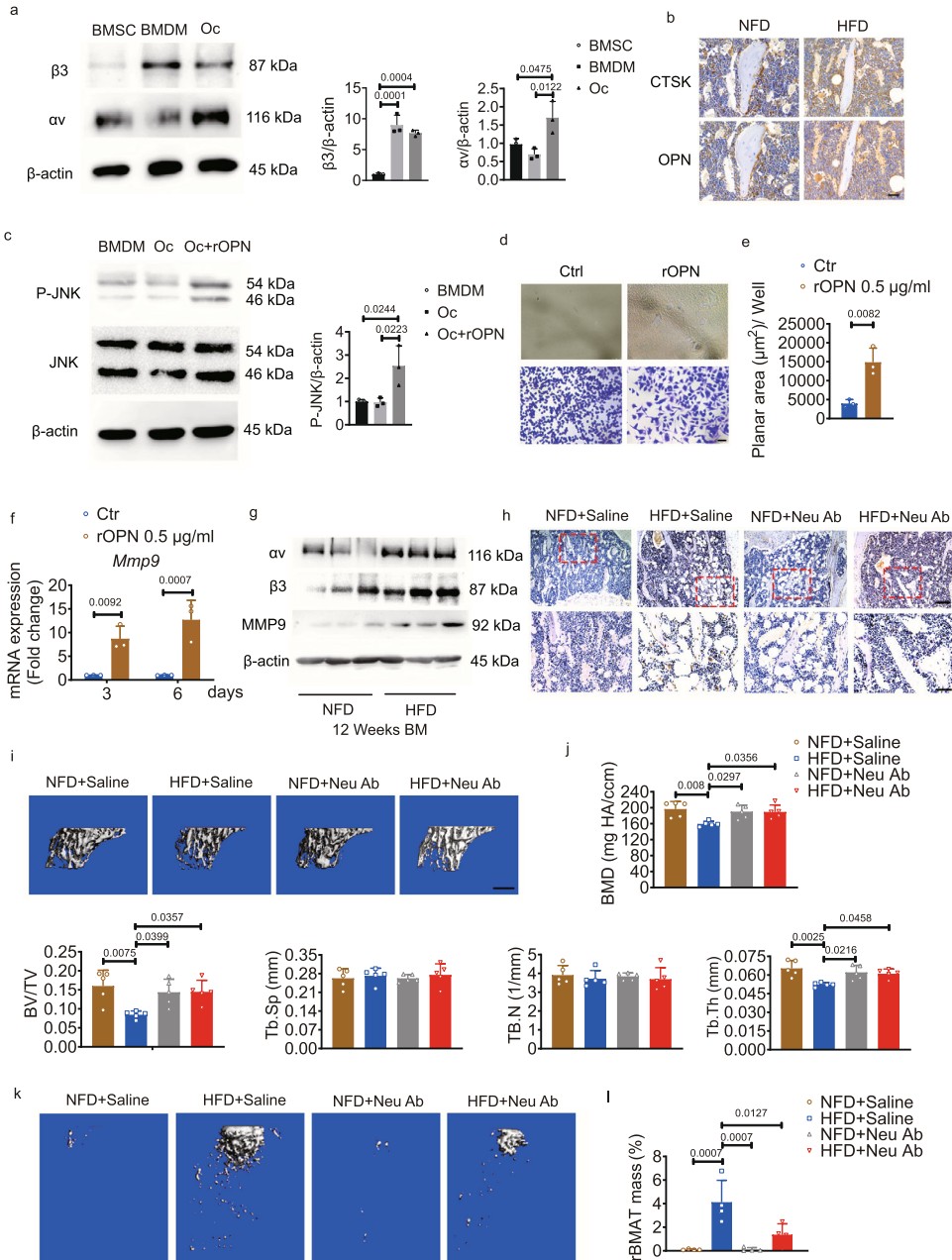

**Fig. 7 eWAT-secreted OPN regulates bone resorption. a** Representative western blots (left) and quantification (right, $n = 3$ biologically independent samples) of αv and β3 in BMSCs, BMDMs, and osteoclasts (Oc) in vitro. **b** Immunohistochemical staining for OPN and cathepsin K (CTSK) in serial sections of tibiae from mice fed for 12 weeks. Scale bar: 50 μm. **c** Representative western blots (left) and quantification (right, $n = 3$ biologically independent samples) of the bands of JNK and p-JNK in BMDMs, Oc, and Oc+rOPN after culture for 4 days in vitro. rOPN, 0.5 μg/ml. **d** Representative images of the osteo-erosion surface (top) and crystal violet staining of the unwashed Osteo Assay surface with or without rOPN treatment under osteoclastogenic induction for 12 days (bottom). Scale bar: 50 μm. **e** Quantification of the bone resorption area from **d** ($n = 3$ biologically independent samples). **f** Relative expression of *Mmp9* in BMDMs with or without rOPN treatment under osteoclastogenic induction for 3 and 6 days, respectively ($n = 3$ biologically independent samples). **g** Representative western blots of αv, β3, and MMP9 in the bone marrow of NFD- and HFD-fed mice at week 12. **h** Immunohistochemical staining of MMP9 from the proximal tibiae of mice after injection of saline or OPN-neutralizing antibody (Neu Ab) into bilateral eWAT for 8 weeks. Scale bar: 100 μm (the bottom row was magnified from the top row, scale bar: 50 μm). **i, j** Representative μCT images (**i**) and quantification ($n = 5$ biologically independent samples) (**j**) of proximal tibiae after mice were injected with saline or Neu Ab. Scale bar: 500 μm. **k, l** Representative μCT images (**k**) and quantification ($n = 4$ biologically independent samples) (**l**) of rBMAT of tibiae after mice were injected with saline or Neu Ab. Scale bar: 500 μm. Images are representative of 3 independent experiments. All data are presented as mean ± SD. One-way ANOVA with Tukey's post-hoc test (**a, c**), two-tailed Welch's t-test (**e**), by two-way ANOVA with Sidak's post-hoc test (**f**), and two-way ANOVA with Tukey's post-hoc test (**j, l**) were used. Source data are provided as a Source Data file. See also Supplementary Tables 7 and 8.

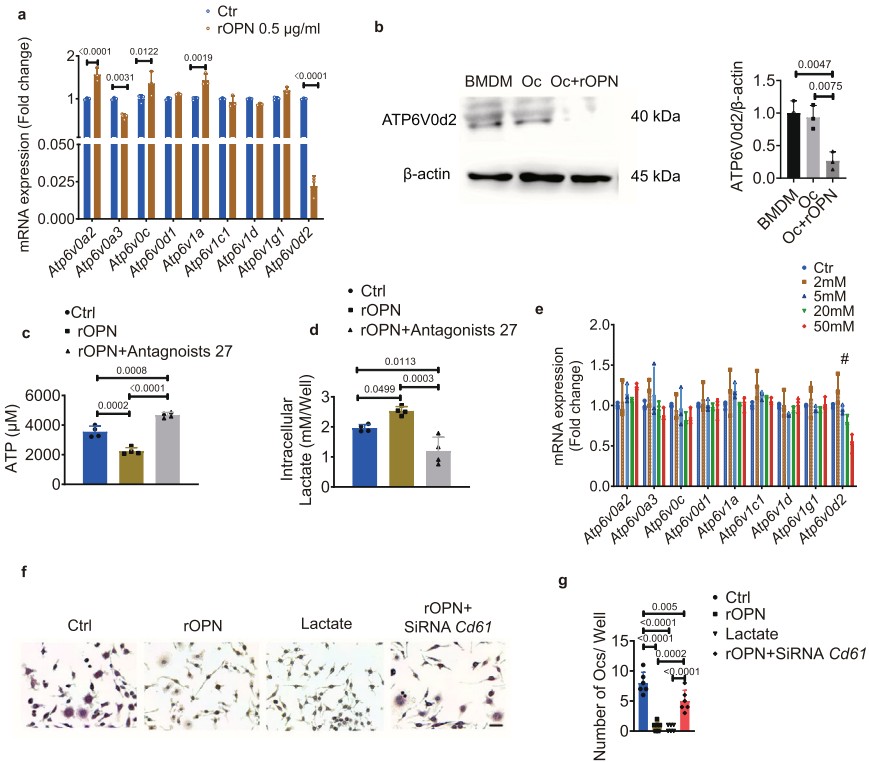

**Fig. 8 Osteopontin regulates bone marrow-derived macrophages through the lactate/ATP6V0d2 axis. a** Relative expression of different *V-ATPase* subunits in BMDMs treated with or without (ctrl) rOPN (0.5 μg/ml) under osteoclastogenic induction for 3 days in vitro (*n* = 3 biologically independent samples). **b** Representative western blot images (left) and quantification (right, *n* = 3 biologically independent samples) of ATP6V0d2 in BMDMs, Oc and rOPN-treated Oc for 4 days in vitro. **c** Total intracellular ATP concentration in BMDMs (ctrl), rOPN (0.5 μg/ml)-treated BMDMs, and BMDMs treated with rOPN (0.5 μg/ml) plus antagonist 27 (100 nM) under osteoclastogenic induction for 0.5 h (*n* = 4 biologically independent samples). **d** Intracellular lactate concentration in BMDMs (ctrl), rOPN (0.5 μg/ml)-treated BMDMs, and BMDMs treated with rOPN (0.5 μg/ml) plus antagonist 27 (100 nM) under osteoclastogenic induction for 10 h (*n* = 4 biologically independent samples). **e** Relative expression of the indicated *V-ATPase* subunits in BMDMs stimulated with a gradient concentration of lactate for 24 h under osteoclastogenic induction. **#** represents significantly decreased (*P* < 0.0001) expression of *Atp6v0d2* in 50 mM lactate-treated BMDMs compared with untreated BMDMs (*n* = 3 biologically independent samples). **f, g** Representative images of TRAP staining (**f**) and counting of Trap+ osteoclasts (*n* = 6 biologically independent samples) (**g**) after osteoclastogenic induction in the presence or absence (ctrl) of rOPN (0.5 μg/ml), lactate (50 mM), and rOPN with siRNA *Cd61* for 5 days. Scale bar: 50 μm. Images are representative of 3 independent experiments. All data are presented as mean ± SD. Two-way ANOVA with Tukey's post-hoc test (**a**, **e**) and one-way ANOVA with Tukey's post-hoc test (**b–d**, **g**) were used. Source data are provided as a Source Data file.

inhibition of the energy regeneration by ATP6V0d2-dependent lipolysis in BMDMs.

## Discussion

The current study has validated the role of OPN in eWAT with bone disorders in HFD-fed mice. As summarized in Fig. 10, eWAT-secreted OPN induces lipophagocytic mobilization of BMDMs to a lipid-rich pool in the bone marrow, while OPN promotes bone matrix degradation by activating osteoclasts in HFD-fed mice. However, the lactate accumulation from OPN regulation blocks both lipolysis and osteoclastogenesis in BMDMs by limiting the energy regeneration for ATP6V0d2 in lysosomes.

An excessively high-calorie diet causes adipose tissue mass and bone mass to diverge, but growing epidemiological evidence offers the explanation that the metabolic syndrome of visceral obesity is responsible for bone loss[3]. Subcutaneous adipose tissue is less metabolically active than VAT, and it may have greater storage capacity. In a clinical study, abdominal obesity, also known as central obesity, was strongly linked to cardiovascular disease, Alzheimer's disease, and other metabolic disorders[35]. Convincing evidence demonstrates that the adipocytes of VAT depots are much more lipolytically active than the subcutaneous portion and contribute more to a high level of plasma FFA[36].

Our results demonstrate that adipose tissues are crucial for the alterations of bone metabolic disorders based on the time course and biological analyses of this study. Of note, eWAT adipocytes are inclined to saturation without storing any fat in obesity, and enlarged adipocytes cause hypoxia and necrosis along with the formation of CLSs[37]. Infiltrating immune cells, especially ATMs, participate in the clearance of dead adipocytes and lipid trafficking, and the ATMs in eWAT concomitantly develop systemic disorders by secreting cytokines. In our study, the greater number of CLSs and stronger inflammatory response in HFD-fed mice compared with NFD-fed mice strongly suggest that eWAT may contribute to disorders of bone metabolism but that iWAT does not. Surgical removal of bilateral eWAT increased trabecular bone and rBMAT mass in HFD-fed mice, while this procedure failed to normalize the ITT response in the HFD-fed group. These results suggest that inflammation in eWAT, but not iWAT, contributes to bone marrow disorders in HFD-fed mice.

Along with *Spp1*, in the microarray analysis of eWAT taken from lean and obese mice[14], *Adam8* (ADAM metallopeptidase domain 8), *Il-7r* (interleukin 7 receptor), *Stap1* (signal transducing adaptor family member 1), *Ctsk*, and other filtered genes are highly upregulated in eWAT, but they have no such characteristic of remote-control through secretion. We found that the most highly upregulated gene in eWAT was *Spp1* and that eWAT

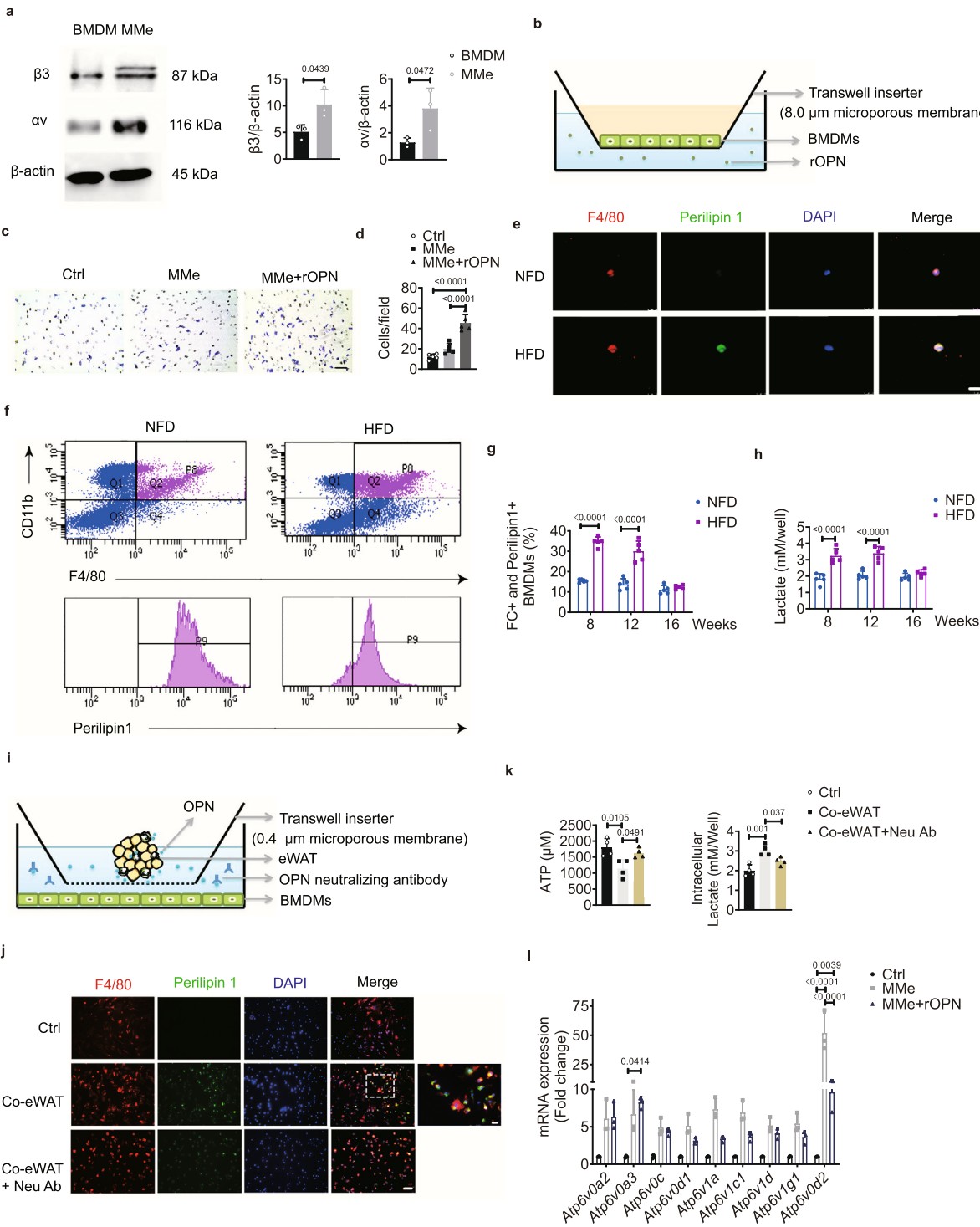

secreted a large amount of OPN into the circulatory system. OPN is expressed by various kinds of cells, including osteoblasts, osteocytes, fibroblasts, dendritic cells, macrophages, hepatocytes, and endothelial cells, and acts systemically after secretion[38]. OPN imparts matricellular phosphoglycoprotein functions as an adaptor and modulator of cell–matrix interactions, favouring cell migration, cell adhesion, and extracellular matrix invasion[3]. OPN plays some role in the lower overall and disease-free survival in patients with several cancers, such as hepatocarcinoma, breast cancer, lung cancer, and prostatic carcinoma, as its level is closely correlated with tumour metastasis, grade, poor prognosis, and early tumour recurrence, largely owing to the soluble OPN that

engages in paracrine signalling at other tissues[39]. The circulatory system can buffer local fluctuations in OPN levels, and systemic changes in its concentration only occur when its main source is removed. We found that removal of bilateral eWAT led to a temporary reduction in OPN in the serum and bone marrow supernatant at week 12 of HFD feeding, indicating that eWAT-secreted OPN accounts for a large proportion of OPN in the circulation and bone marrow. The regulatory role of eWAT-secreted OPN in bone is highlighted by the fact that modified OPN (rOPN-FITC) targets trabecular bone. More importantly, the progression of bone loss is slowed after clearance of OPN from eWAT by local injection of Neu Ab.

**Fig. 9 OPN induces lysosomal-dependent lipid metabolism in BMDMs. a** Representative western blots (left) and quantification (right, $n = 3$ biologically independent samples) of OPN receptor αv and β3 expression in BMDMs and metabolically activated macrophages (MMes) after culture for 4 days in vitro. **b** Schematic diagram illustrating the co-culture system as indicated below (**c**). **c**, **d** Representative images (**c**) and quantification ($n = 5$ biologically independent samples) (**d**) of crystal violet staining of BMDMs (ctrl) and MMes treated with or without rOPN for 24 h in a Transwell system. Scale bar: 50 µm. **e** Immunofluorescence images of buoyant BMDMs (F4/80) containing lipid vesicles (Perilipin 1) isolated from bone marrow supernatant at week 12. Scale bar: 15 µm. **f** FACS analysis of FC + (F4/80+ and CD11b+ , top) and Perilipin1+ (bottom) BMDMs from NFD- (left) and HFD-fed (right) mice. P9, FC + and Perilipin1+ BMDM population from Q2. **g** Percentage of FC + and Perilipin1+ BMDMs ($n = 5$ biologically independent samples). **h** Lactate concentration in bone marrow supernatant ($n = 5$ biologically independent samples). **i** Schematic diagram illustrating the co-culture system as described below (**j**). **j** Immunofluorescence staining of the co-cultured BMDMs for 4 days in vitro; the eWAT from mice fed the HFD for 8 weeks was placed in the upper chamber; Neu Ab (2.0 µg/ml) was added into the lower chamber. Scale bar: 50 µm. The right-most image is an enlarged view of the field in the white dashed box. Scale bar: 12.5 µm. **k** Total intracellular ATP concentration (left, co-culture for 0.5 h) and lactate concentration (right, co-culture for 10 h) in BMDMs with the treatments indicated in **j** ($n = 4$ biologically independent samples). **l** Relative expression of different *V-ATPase* subunits in ctrl (BMDMs), MMes, and MMes+ rOPN after culture for 24 h ($n = 3$ biologically independent samples). rOPN, 0.5 µg/ml. Images are representative of 3 independent experiments. All data are presented as mean ± SD. Two-tailed Welch's t-test (**a**), one-way ANOVA with Tukey's post-hoc test (**d**, **k**), two-way ANOVA with Sidak's post-hoc test (**g**, **h**), and two-way ANOVA with Tukey's post-hoc test (**l**) were used. Source data are provided as a Source Data file.

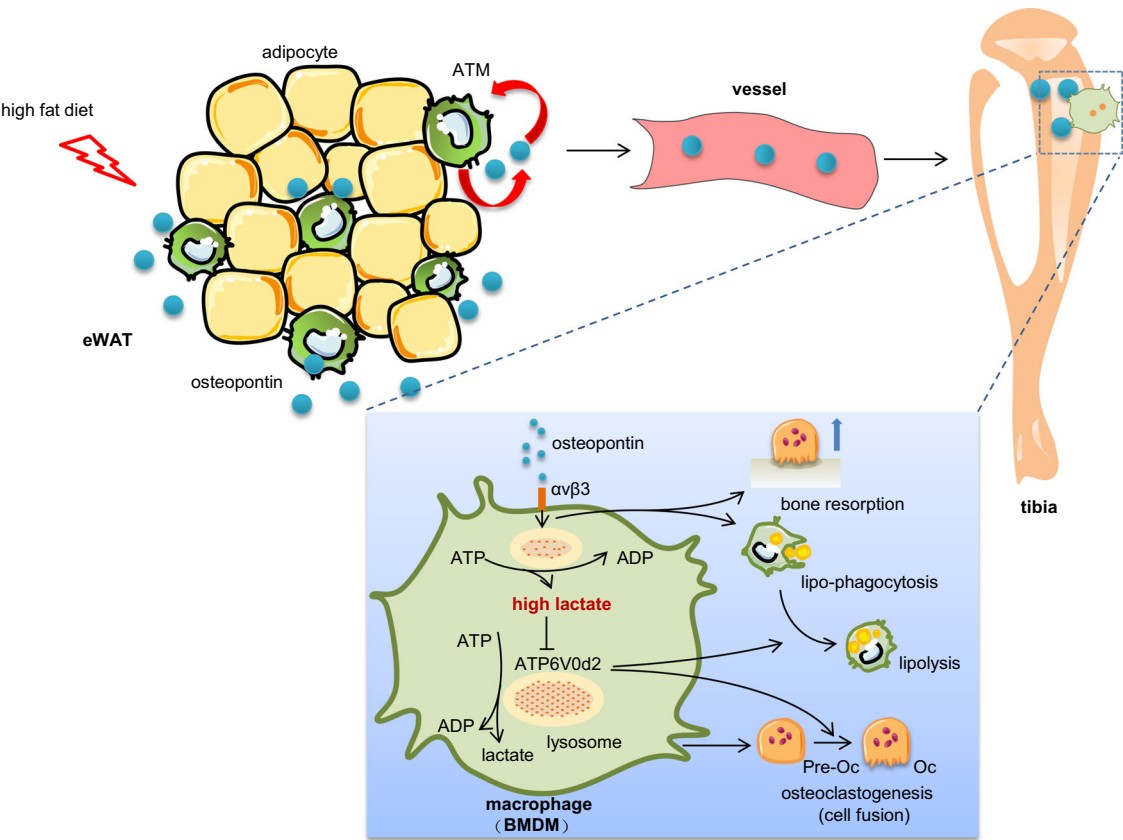

**Fig. 10 Schematic diagram showing that macrophages in epididymal adipose tissue secrete OPN to regulate bone homeostasis.** eWAT-secreted OPN accumulates in the bone marrow compartment where OPN promotes bone resorption and lipophagocytic mobilization, but the excess accumulation of lactate inhibits pre-osteoclast fusion and neutral lipid hydrolysis via lysosomal-dependent ATP6V0d2 in BMDMs.

A conventional paradigm of rBMAT biology is that its adipocytes are strikingly dynamic and that their mass increases with diet-induced obesity, steroid treatment, alcohol drinking, or ageing[22]. However, we have uncovered an unusual alteration of the rBMAT depot that contrasts with the conventional concept that rBMAT continuously accumulates with diet-induced obesity. Due to the larger size of the adipocytes, ATMs prefer to form a tight attachment (a lysosomal synapse) to dead adipocytes[40]. They acidify the contact region using the plasma membrane proton-pumping V-ATPase and secrete lysosomal contents into the lysosomal synapse[40]. Lysosomal lipases in the acidic milieu liberate FFAs, which are subsequently internalized by macrophages, leading to lipid recycling[40], while lipid-filled exosomes are a source of lipids for local macrophages, and exosomes contain the specific component perilipin 1[41]. We found that lipid-laden BMDMs were also positive for perilipin 1, demonstrating that BMDMs, having ATM-like features, are involved in lipophagocytosis in the bone marrow. The inhibition of lysosome function in BMDMs compromises lipid degradation by increasing their accumulation of intracellular lipids, suggesting that BMDMs are important mediators of lipid recycling in bone marrow, like ATMs in eWAT. These results may explain why the lipid-containing BMDM percentage tended to be consistent with that of the local FFA level in the bone marrow, which initially increased and then decreased at week 16 after feeding with HFD.

Cells poorly replenish ATP following surges in energy demand, suggesting that macrophages with high bioactivity easily disturb the dynamic ATP recycling needed for downstream signalling. Indeed, the ATP-dependent process of pre-osteoclast fusion or neutral lipid hydrolysis via lysosomes is restrained by the accumulation of glycolysis products, such as lactate, produced during OPN-regulated bone resorption or lipophagocytic mobilization. These findings suggest that OPN exerts forward effects on lipid recycling and bone resorption but synchronously experiences feedback inhibition of these processes from glycolytic lactate.

Modulation of OPN expression might be one way to alleviate bone metabolic disorders based on our present understanding of the roles of OPN. In post-menopausal women with diabetes, OPN might be a useful marker for bone fracture and a worsening lipid profile[42]. Our results provide insight into the role of eWAT-secreted OPN in bone homeostasis during obesity, highlighting the potential of OPN-targeting strategies to prevent bone loss under HFD feeding. Therefore, applying the injectable hydrogels reported in our previous investigation[43] in combination with *spp1* siRNA or OPN-neutralizing antibody may be a promising therapy to enhance bone fracture healing via temporospatial intervention in obese individuals. Biodegradable membranes or scaffolds that can immobilize OPN-neutralizing antibodies or deliver OPN receptor antagonists, such as integrin antagonist 27, may have broad applications for improving bone regeneration in conditions of fracture nonunion or bone defects.

The current study neither eliminates the possibility that OPN from other sources of deposits (for example, the liver and kidney) nor other cytokines (for example, Il-1β and TNF-α) are involved in the bone marrow metabolism effects we saw. Although the secretion of OPN contributes to these inflammatory responses and lipid trafficking in eWAT, the initial factor driving the homeostatic disorder of eWAT remains elusive.

In summary, our study shows that eWAT-secreted OPN accumulates in the bone marrow compartment, where it promotes bone resorption and lipophagocytic mobilization, but the excess accumulation of lactate inhibits pre-osteoclast fusion and neutral lipid hydrolysis via the lysosomal-dependent ATP6V0d2 in BMDMs. Surgical removal of eWAT, local injection of clodronate liposomes, or local injection of OPN-neutralizing antibody showed comparable amelioration of HFD-induced bone loss in mice. These results suggest an avenue for the development of therapeutic strategies to modulate visceral adipose tissue inflammation and thus mitigate obesity-related bone loss.

## Methods

**Animals**. Twelve-week-old male *C57BL/6* mice were fed a high-fat diet (HFD; 60% of total calories from fat, 20% from carbohydrate and 20% from protein, cat# D12492, Research Diets Inc.)[25]. The control group were given a normal-fat diet (NFD; 10% of total calories from fat, 70% from carbohydrate, and 20% from protein, cat# D12450J, Research Diets Inc.). Body weight was measured weekly. If animals showed signs of suffering during the observation period, they were immediately humanely euthanized. All animals were housed under a 12 h light/dark cycle, 18–23 °C ambient temperature, and 70% humidity at the Experimental Animal Center at the Prince of Wales Hospital in Hong Kong and received NFD/HFD and water *ad libitum*. The specified experimental protocols were approved by the Animal Experiment Ethics Committee of the Chinese University of Hong Kong (17-184-MIS-5-C, 19-036-MIS-5-C, 20-111-MIS-5-B, and 21-027-MIS).

**Micro-computed tomography (μCT) imaging**. The microarchitecture was scanned by a μCT-40 imaging system (Scanco Medical, Brüttisellen, Switzerland), and analysis of trabecular bone under the growth plate of a 1.5 mm length of the proximal tibia was obtained. The analysis covered bone mineral density (BMD), bone volume fraction (BV/TV), trabecular bone thickness (Tb.Th), trabecular bone separation (Tb.Sp), and trabecular bone number (Tb.N)[44]. For analysis of bone marrow adipose tissue (BMAT), tibiae were harvested by careful removal of soft tissues and fixed in 4% paraformaldehyde (PFA) for 24 h. Tibiae were decalcified in 12.5% ethylenediaminetetraacetic acid (EDTA, pH 7.4, Sigma, cat# 798681) for 2 weeks. Decalcified tibiae were immersed in glass tubes containing a 1:1 ratio of 4% osmium tetroxide (Sigma, cat# 75632) and 5% potassium dichromate (Sigma,

cat# P2588) solution to stain the BMAT for 2 days at room temperature. BV/TV was measured in the proximal tibia (a 3 mm length) to calculate rBMAT mass and in the distal tibia (a 3 mm length) to calculate cBMAT mass[21]. The reconstruction thresholding value was optimized to encapsulate the target image.

**Enzyme-linked immunosorbent assay (ELISA)**. ELISA was conducted on serum, conditioned medium, and bone marrow supernatant to detect OPN (Abcam, cat# ab100734), FFA (Abcam, cat# ab65341), or lactate (Abcam, cat# 65331)[45]. The concentrations of OPN were measured by diluting serum 300-fold, bone marrow supernatant 100-fold, eWAT conditioned medium 100-fold, and iWAT conditioned medium 20-fold. Bone marrow supernatant was diluted 1.25-fold to measure the concentration of FFA.

**Histological analysis**. To isolate bones for histological analysis, mice were first euthanized. The isolated bones were fixed in 4% PFA for 24 h. Tibiae were decalcified in 12.5% EDTA for 14 days, embedded in paraffin, and sectioned to 5 μm. Adipose tissues and other viscera were fixed in 4% PFA for 24 h, embedded in paraffin, and sectioned to 5 μm. Paraffin sections were stained according to a routine haematoxylin and eosin (H&E) staining protocol; immunohistochemistry for OPN was performed (Invitrogen, cat# PA125152; Abcam, cat# ab8448) and CTSK (Abcam, cat# 37259, 1:200) according to a previously established protocol[46]. Briefly, sections were deparaffinized by xylene, immersed in 3% hydrogen peroxide for 15 min in the dark, and processed with citrate buffer (Dako, cat# 2369, pH 6.0) for 20 min at 80 °C. Primary antibody was diluted in blocking buffer (1% BSA and 0.1% Triton X-100 in PBS) overnight at 4 °C, and diluted secondary antibody in blocking buffer was applied for 2 h at room temperature. Omitting the primary antibody was used as a negative control. Immunofluorescence staining for OPN (1:200), F4/80 (Invitrogen, cat# 11-4801-82, 1:200), and Perilipin1 (Abcam, cat# 3526, 1:200) was performed. Briefly, sections were deparaffinized by xylene and subjected to an antigen retrieval procedure using citrate buffer at 80 °C for 20 min. The sections were then incubated with primary antibody overnight at 4 °C. Next, they were incubated with a secondary antibody for 1 h at room temperature in the dark[47]. Images were digitalized with a microscopic imaging system (Leica DM5500; Leica Micro-systems, Wetzlar, Germany).

**Insulin tolerance test**. After fasting for 6 h, the mice were intraperitoneally injected with insulin (0.70 unit/kg body weight, Sigma, cat# I9278). The glucose concentration was measured in blood taken from the tail vein at 30 min intervals from 0 to 120 min[48].

**Targeted gene prediction from the gene chip microarray assay**. We used a gene chip generated by another research group[14] to demonstrate the effects of diet on the gene expression of BMAT and eWAT in mice. Microarray data files were acquired from the Gene Expression Omnibus with the accession number GSE27017.

**Conditioned medium**. Mice were sacrificed to collect same-weight eWAT and iWAT and separately cultured in 2 ml fetal bovine serum (FBS, Gibco, cat# 16140071)-free medium (Dulbecco's modified Eagle medium, DMEM) for 24 h at 37 °C and 5% CO₂. All procedures were done under aseptic conditions. The OPN level in the conditioned medium was measured by ELISA.

**Co-culture assays**. For the co-culture of BMDMs with eWAT, BMDMs were seeded at a concentration of $2.5 \times 10^5$ cells/ml in the lower chamber of a 12-well plate with DMEM/F12 medium containing M-CSF (30 ng/ml, Gibco, cat# PHG6054), 10% FBS, and 1% PSN in a 37 °C incubator with 5% CO₂. A Transwell insert (0.4 μm polycarbonate filter, Corning, cat# CLS3401-48EA) was placed as the upper chamber. eWAT (0.5 g) was isolated from HFD-fed mice (at week 8) and cultured in the upper chamber, which shared medium with the lower chamber. Neu Ab (2.0 μg/ml, Invitrogen, cat# PA125152) was added to the lower chamber medium. The medium was carefully changed every second day[49]. All procedures were maintained under aseptic conditions.

**Isolation of bone marrow supernatant, cells, and bone marrow protein**. Both tibiae were harvested directly after euthanasia and placed in 0.5 ml microtubes, which had their bottom cut off and were inserted into 1.5 ml Eppendorf tubes. These nested tubes with bones were spun for 9 s at 13,000 × g to acquire bone marrow pellets. The pellets were resuspended in 500 μl erythrocyte lysis buffer (Thermo Fisher Scientific, cat# 00-4333-57) and centrifuged at 500 × g for 3 min at 37 °C. The top layer of lipid-containing cells was collected as the buoyant adipocyte-rich fraction. The separated middle layer was collected as a bone marrow supernatant for ELISA[25]. The pellets were washed with phosphate-buffered saline (PBS) and stained with fluorescence-tagged antibodies to detect cell lineages. For the extraction of whole-bone-marrow protein, the pellets described above were resuspended in 500 μl radioimmunoprecipitation assay (RIPA) lysis buffer (Cell Signaling Technology, cat# 9806) containing protease and phosphatase inhibitors and further processed as described below.

**Tracing FITC-labelled recombinant human OPN and FITC-labelled OPN-neutralizing antibody**. Before adding recombinant human OPN (rOPN, Abcam, cat# ab92964) into the FITC mix (Abcam, cat# ab188285), 1 μl of modifier chemical was added to 10 μl rOPN (0.15 mg/ml), and it was slightly agitated. After incubation for 30 min, 1 μl quencher chemical was added to 11 μl of rOPN (with added modifier reagent), and this was lightly agitated to mix. rOPN conjugated with FITC (rOPN-FITC) was incubated at room temperature for 15 min in the dark before use. Mice fed a HFD for 12 weeks were anaesthetized (ketamine (60 mg/kg, i.p.) and xylazine (4 mg/kg, i.p.)) and shaved at the mid-ventral abdomen. rOPN-FITC was injected into the unilateral (left) eWAT. To visualize the injected rOPN-FITC reagents, mice were euthanized at either 1 h or 24 h post-injection, and tissues and organs were isolated and imaged by an IVIS200 imaging system (Xenogen Imaging Technologies, Alameda, CA, USA)[50].

Before adding Neu Ab to the FITC mix, 10 μl of modifier chemical was added to 100 μl of Neu Ab (0.1 mg/ml) and slightly agitated. After incubation for 30 min, 10 μl quencher chemical was added into 110 μl of Neu Ab (with added modifier reagent), which was slightly agitated to mix. The Neu Ab conjugated with FITC (Neu Ab-FITC) was incubated at room temperature for 15 min in the dark before use. Mice fed an HFD for 12 weeks were anaesthetized and shaved at the mid-ventral abdomen. Neu Ab-FITC was injected into unilateral (left) eWAT. To visualize the injected Neu Ab-FITC reagents, mice were euthanized at either 1 h or 24 h post-injection to isolate relevant tissues and organs for imaging acquisition by an IVIS200 imaging system.

**Surgical removal of epididymal white adipose tissue (eWATx) and perirenal white adipose tissue (PATx)**. Twelve-week-old mice were anaesthetized with ketamine (60 mg/kg, i.p.) and xylazine (4 mg/kg, i.p.) for all surgical procedures. All procedures were done under aseptic. Mice were placed on a warming operating table to maintain body temperature. The abdominal hair was shaved, and the skin was treated with betadine. A small mid-ventral abdominal incision was made to expose the bilateral eWAT. Bilateral eWAT was surgically removed without damaging the blood supply to the reproductive organs (note: this is vital for bone metabolism[12]). For the sham group, bilateral eWAT was pulled out and placed back through the same abdominal incision. Gentocin was subcutaneously administered once daily at 5 mg/kg for 3 days after surgery as an antibiotic. After a normal diet for 3 days, mice continued on the NFD or HFD until their experimental time point was reached.

For the PATx, 12-week-old mice were anaesthetized and prepared for surgery. A ventral incision into the abdomen was made to expose the left kidney, and the PAT was excised without damaging the blood supply. The same procedures were performed for the other PAT. In the sham group, bilateral PAT depots were exposed through the same abdominal incisions. Gentocin was subcutaneously administered once daily at 5 mg/kg for 3 days after surgery. After a normal diet for 3 days, mice continued on the NFD or HFD until euthanasia at the defined sampling times.

**Isolation of SVF, AF, and ATMs**. To isolate the SVF and AF from eWAT and iWAT[25], eWAT and iWAT were separately minced and incubated in HEPES buffer DMEM (Gibco, cat# 12430054) containing 10 mg/ml fatty acid-poor bovine serum albumin (BSA). After centrifugation at $500 \times g$ for 10 min, the pellets were incubated in Liberase$^{TM}$ (0.14 units/ml, Roche, cat# 5401020001) for 30 min at 37 °C. The floc at the bottom was regarded as the SVF, and the floating cells were regarded as the AF. The AF was digested for another 30 min with Liberase$^{TM}$ and suspended in DMEM containing 10% FBS. The pellets and floating cells were collected after centrifugation at $500 \times g$ for 10 min at room temperature. These digestive steps were repeated until no precipitation occurred. The SVF was incubated in erythrocyte lysis buffer for 3 min at room temperature. The SVF and AF were resuspended in TRIzol reagent (Invitrogen, cat# 15596026) to analyze mRNA expression.

**Flow cytometry analysis**. ATMs and BMDMs were stained with fluorescence-tagged antibodies to detect cell lineages. To separate the ATMs, the erythrocyte-deleted SVF was passed through a 70 μm nylon mesh (Corning, cat# CLS431751-50EA) and then washed with PBS. SVF was stained and screened by flow cytometry using anti-CD11b (Thermo Fisher Scientific, cat# 53-0112-80, 1:100) and anti-F4/80 antibodies (Thermo Fisher Scientific, cat# 12-4801-80, 1:100). Single-colour controls were used to set compensations, and fluorescence minus one control was used to set the gate. The isolated ATMs were resuspended in TRIzol reagent for mRNA expression analysis.

BMDMs were passed through a 70 μm nylon mesh as described above and washed with PBS. BMDMs were stained with antibodies against F4/80 (1:80), CD11b (1:80), and perilipin1 (Abcam, cat# ab3526, 1:80)[51]. Single-colour controls were used to set compensations, and fluorescence minus one control was used to set the gate. Data were analyzed using BD FACSDiva software v8.0.1.

**Transwell assay**. BMDMs were seeded on the upper layer of the culture insert with an 8.0 μm permeable membrane (Corning, cat# 353097), and the medium containing the test reagent was placed in the lower chamber[31]. At the experimental time point, the migrated BMDMs were fixed with 4% PFA for 2 min, permeabilized

with absolute methanol for 20 min, stained with crystal violet (0.5%, Sigma, cat# C0775) for 10 min, and photographed under a microscope (ZEISS AxioPlan2, Germany) and counted in ImageJ (version 1.52 v, USA).

**Isolation of bone marrow mesenchymal stem cells (BMSCs)**. For BMSC isolation, bone marrow cells from male *C57BL/6* mice (6 weeks old) were flushed out with minimum essential medium (MEM) without FBS[52]. Erythrocyte-deleted BMSCs were cultured in MEM (Gibco, cat# 61100061) supplemented with 15% FBS and 1% penicillin-streptomycin-neomycin (PSN, Gibco, cat# 15640055) and incubated at 37 °C with 5% $CO_2$. The medium was changed every 3 days. Cells were passaged until 80% confluent.

**Isolation of bone marrow cells for mRNA expression analysis**. Fresh bone marrow from proximal tibiae was flushed out with PBS containing 1% fatty acid-free BSA and 1% RNase and DNase-free water using a 25 gauge needle[14]. Red blood cells were lysed using erythrocyte lysis buffer for 3 min at 4 °C. After washing them with PBS and centrifuging them at $500 \times g$ for 5 min, cells were added to TRIzol reagent.

**Osteoclastogenic differentiation and lipid metabolic activation**. To isolate BMDMs, the bone marrow from male *C57BL/6* mice (6 weeks old) was flushed out with MEM and plated in MEM containing 10% FBS and 1% PSN. After cultured for 12 h, non-adherent cells were collected and centrifuged for 5 min at $300 \times g$. The precipitated pellet was resuspended in DMEM/F12 (Gibco, cat# 10565018) medium containing 10% FBS, 1% PSN, and M-CSF (30 ng/ml, Gibco, cat# PHG6054) and incubated at 37 °C with 5% $CO_2$.

BMDMs at passage 1 were cultured in DMEM/F12 medium containing 10% FBS, 1% PSN, M-CSF (30 ng/ml), and RANK ligand recombinant human protein (RANKL, 30 ng/ml, Gibco, cat# PHP0034) for 2–3 days to generate pre-osteoclasts and 4–5 days to generate multinucleated mature osteoclasts[53].

To generate MMes, BMDMs were cultured in DMEM/F12 medium containing 10% FBS, 1% PSN, glucose (30 mM, Sigma, cat# 47829), insulin (10 nM, Sigma, cat# I2643), and palmitate (0.4 mM, Sigma, cat# P9767) for 24 h to imitate the MMe micro-milieu in vitro[54]. Palmitic acid powder (Sigma, cat# P9767) was added to a 10% solution of fatty acid-free BSA and was dissolved by shaking gently overnight at 37 °C to yield the stock solution (8 mM). BSA alone (0.5%, w/v) was added in the control group[55].

**Bone resorption assay**. BMDMs were cultured on Corning Osteo Assay surface plates (Corning, cat# CLS3987) with osteoclastogenic induction medium (10% FBS, 1% PSN, M-CSF (30 ng/ml), and RANKL (30 ng/ml)). After incubation for 12 days with or without rOPN (0.5 μg/ml, Abcam, cat# ab92964), the cells were removed using 4% NaClO, and then the plate was washed with deionized water[56]. Pits were observed as bright spots on the slides under a microscope (ZEISS AxioPlan2, Germany), and the bone erosion area was calculated within ImageJ (version 1.52 v, USA).

**ATM depletion**. The abdominal hair was shaved, and the skin was treated with betadine. ATMs were depleted by injecting clodronate liposomes (0.675 mg/unilateral/3 days, LIPOSOMA, cat# C10E0218) into the bilateral eWAT under inhalation isoflurane without surgical incision every 3 days from 12 weeks of age. Mice were simultaneously fed an NFD or HFD for 8 weeks[57]. The control group mice received the control liposome (LIPOSOMA, cat# C14E0218).

**Injection of Neu Ab antibody**. The abdominal hair of the mice was shaved, and the skin was treated with betadine. OPN was depleted by injecting Neu Ab (1.25 μg/unilateral/3 days) into the bilateral eWAT under inhalation isoflurane anaesthesia without surgical incision every 3 days in 12-week-old mice. Mice were simultaneously fed a NFD or HFD for 8 weeks[58]. The control group mice received an injection of saline (25 μl/unilateral/3 days).

**Osteopontin-receptor integrin β3 transfection in vitro**. The small interfering RNA (siRNA, Thermo Fisher Scientific, cat# AM16708) against *integrin β3* (*Cd61*) was mixed and diluted in Opti-MEM (Gibco, cat# 31985088) to a final concentration of 100 nM. BMDMs were transiently transfected using Lipofectamine RNAiMAX Transfection Reagent (Thermo Fisher Scientific, cat# 13778-075) according to the manufacturer's instructions. Control BMDMs were transfected with negative control siRNA (Thermo Fisher Scientific, cat# AM4611). Transfection efficiency was assessed by measuring mRNA expression 24 h after transfection.

**ATP and lactate assessment**. Experiments were set up in 96-well plates ($1 \times 10^4$ cells/well), in which the ATP level was measured after the indicated treatments for 30 min, according to the manufacturer's instructions (Abcam, cat# ab113849). Intracellular ATP content was measured by lysing cells with substrate solution and measuring the luminescence with a luminometer (VICTOR$^{TM}$ X4, Perkin Elmer, USA).

To assess the intracellular lactate content, treated cells were harvested and resuspended in lactate assay buffer. To assess the bone marrow lactate content, a bone marrow supernatant was collected. Lactate concentration levels were measured enzymatically according to the manufacturer's instructions (Abcam, cat# ab65331).

**Western blot analysis.** Tissues or cells were totally lysed in RIPA buffer with protease and phosphatase inhibitors (Roche). Protein extracts were electrophoresed through sodium dodecyl sulfate-polyacrylamide gels and transferred to poly-vinylidene fluoride membranes[59]. Membranes were blocked with blocking buffer (5% dry milk) in Tris-buffered saline with 0.1% Tween-20 for 60 min at room temperature and incubated with primary antibodies (Integrin αv, Abcam, cat# ab179475, 1:2000; Integrin β3, Abcam, cat# ab38460, 1:1000; β-actin, Cell Signaling Technology, cat# 4970 s, 1:1000; ATP6V0d2, Bioss, cat# bs-12548R, 1:1000; MMP9, Abcam, cat# ab38898, 1:1000; JNK, Cell Signaling Technology, cat# 9252 s, 1:1,1000; P-JNK, Cell Signaling Technology, cat# 9255 s, 1:1,1000) diluted in blocking buffer overnight at 4 °C. These membranes were incubated with secondary antibodies, and hybrid HRP was detected using Pierce™ ECL Western Blotting Substrate (Thermo Fisher Scientific, cat# 32106) and quantified with ImageJ (version 1.52 v). Western blot data in the figures are representative of 3 or more independent experiments. The uncropped and unprocessed scans of the blots are provided in the Source Data file.

**Quantitative real-time polymerase chain reaction (qRT-PCR).** Total RNA was extracted using TRIzol reagent (Invitrogen, cat# 15596026). RNA was quantified spectrophotometrically based on the A260 value in an ND-2000 spectrophotometer (NanoDrop Technologies, Wilmington, DE, USA). mRNA was reverse-transcribed to cDNA using a cDNA kit (Takara). cDNA was amplified in a mixture reaction system containing SYBR Green qPCR SuperMix-UDG (Takara) and specific primer sequences (Supplementary Table 1). The qRT-PCRs were run on a Quant-Studio™ 12 K Flex Real-time PCR system (Life Technologies, Thermo Fisher Scientific).

**Statistical analysis.** All mice were randomly assigned to the indicated groups. Although all treatments and assessments were not blinded, the results were confirmed by three independent investigators. For all assays, the sample size corresponded to individual mice or cells that received treatment instead of the average of technical duplicates. Statistical analysis was performed using the unpaired two-tailed Welch's $t$-test for between-group comparisons and one-way ANOVA with Tukey's post-hoc test or two-way ANOVA with Tukey's post-hoc test for between-group comparisons. This is also specified in the figure legends. For the longitudinal measures (for example Figs. 1, 2, and S2), mixed-effect or two-way ANOVA with repeated measures was conducted, followed by Sidak's post-hoc test (as indicated in the figure legends). Numerical values for each measurement are displayed as mean ± standard deviation (SD). $P < 0.05$ was defined as statistically significant. GraphPad Prism (version 8.2.1) was used for all statistical analyses.

**Reporting summary.** Further information on research design is available in the Nature Research Reporting Summary linked to this article.

## Data availability

Source data are provided with this paper as a Source Data file, including all data which support the findings of the current study. All other information is available within the manuscript, supplementary information file, or by reasonable request from the corresponding author. Source data are provided with this paper.

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

## Acknowledgements

This work was supported in part by Theme-based Research Scheme of Hong Kong (No. T13-402/17 N), Health and Medical Research Fund of Hong Kong (18190481), Areas of Excellence (AoE/M-402/20), National Natural Science Foundation of China (81802152), and Natural Science Foundation of Guangdong Province (2019A1515012224).

## Author contributions

B.D., J.X., and L.Q.: conception and design; B.D., J.X., X.L., L.H., H.W., H.Y., J.M., L.Z., J.W., W.T., D.H.C., Ye.L., X.H., P.H., Z.C., H.Z., Y.L., Y.Y., Q.J., and L.Q.: experiments and data analysis; B.D., J.X., C.H., X.L., and L.Q.: article writing with contributions from other authors.

## Competing interests

The authors declare no competing interests.
