## [Peer Review File · Nature Communications]

REVIEWER COMMENTS

Reviewer #1 (Remarks to the Author):

In the paper entitiled “Macrophages in Epididymal Adipose Tissue Secrete Osteopontin to Regulate Bone 1 Homeostasis” Dai and colleges showed, that that macrophages in eWAT are the main source of osteopontin, which selectively circulates to the bone marrow and promotes the degradation of bone matrix by activating osteoclasts as well as modulates bone marrow-derived macrophages (BMDMs) to engulf the lipid droplets released from adipocytes within the bone marrow in mice. The Authors showed, that surgical removal of eWAT, local injection of clodronate liposome (for depleting macrophages), and local injection of osteopontin neutralizing antibody show comparable effect on ameliorating HFD-induced bone loss in mice. Presented by the Authors data suggest a future avenue for the development of novel therapeutic strategies to ameliorate obesity-related bone loss. Presented for review Article are well designed and well done. There are major correction necessary including:

1. Explaining small gropus of animals included into the research?
2. Explaining why in 12 week there are big secretion of OPN?
3. Please rebuild introduction and focuse more on OPN and bone density correlation
4. Please in discussion section highlight the potential therapeutic strategy for healing the bone injuries by means of modulation of OPN level

Reviewer #2 (Remarks to the Author):

In this study, Dai et al. have explored the relationship between adipose tissue mass and bone homeostasis. Through studies in mice, they have identified osteopontin to be a protein secreted by macrophages in epididymal white adipose tissue that promotes bone matrix degradation. While there are interesting findings presented in the manuscript, important aspects are missing and the resulting picture therefore feels incomplete.

Major points:

1. As injections with clodronate liposomes have been shown to impact on multiple metabolic processes in mice, e.g. alter insulin sensitivity and protect against diet-induced obesity, additional experiments are required to prove that osteopontin produced by visceral adipose tissue macrophages induce bone remodeling. Could the authors specifically deplete osteopontin from visceral adipose tissue macrophages (as has been done elsewhere, PMID: 23630254) and then measure effects on bone remodeling?
2. There is no difference in circulating osteopontin levels comparing mice fed a normal or high-fat diet. Still, the authors claim that increased expression of osteopontin during weight gain allows crosstalk between fat and bone. It is suggested that osteopontin receptor levels are key to this, but there is no difference comparing immunohistochemical staining of OPN bound to cells in proximal tibiae of NFD and HFD-fed groups. Can the authors provide evidence on exactly how this works?

Reviewer #3 (Remarks to the Author):

The authors present an interesting study that reaffirms and extends the notion that macrophage recruitment to visceral white adipose tissue depots contributes to the negative effects of visceral obesity on bone. They show that this effect is mediated primarily by the release of circulating osteopontin by the macrophages within the gonadal fat depot, subsequently inducing a cascade of events within the bone and bone marrow that impact osteoclast function and osteoclastogenesis. This manuscript builds upon their previous work and other studies in the field showing that osteopontin is a critical mediator of high-fat diet induced adverse phenotypes. Currently, there are key issues with the data that need to be considered. However, if addressed, this study would be an excellent addition to the literature.

Major Concerns:

1. Figure 1. Why does the eWAT and rBMAT recover at 16-weeks? Did you happen to have a group of non-responders (selection bias) or is this recovery despite ongoing HFD consumption a repeatable result?
2. Figure 2. What about other visceral WAT depots? In rodents, the perirenal WAT is a large visceral depot that more closely replicates visceral WAT in humans (humans do not have an eWAT depot). There are also adipocytes present around the omental tissues.
3. Figure 3. Does eWAT removal actually rescue bone? The current results are relatively inconclusive and underpowered. At 8-weeks the answer is clearly 'no'. At 12-weeks there is a slight improvement in BV/TV, but it does not appear that the 'eWATx' x HFD interaction effect would be significant by ANOVA. As this is the core observation for rest of the paper – it should be strengthened/repeated to

confirm or deny the result. The authors could also consider removing the other visceral fat (ex. perirenal fat) if this is found to be a source of osteopontin.

4. Figure 4. Missing NFD + clodronate liposome control group. Do clodronate liposomes impact bone in the NFD group, independent of HFD? Please show individual data points (as in Fig.3).

5. Figure 5. The presented immunostaining results look like autofluorescence of the entire adipocyte extracellular matrix. Additional validation of the antibody with positive and negative controls is needed. It is agreed that given the large increase in mRNA, corresponding increases in protein are likely.

6. Figure 6. Same comments about OPN IHC. The bone images provided are not suitable to localize the staining and appropriate positive and negative controls are needed. In addition, what happens to Spp1 expression in the floating fraction in 6B? In addition to adipocytes, this fraction would presumably contain any lipid-laden macrophages.

7. Figure 7. Missing control diet + saline and control diet + Neu Ab control groups. Does the OPN neutralizing antibody have the same positive effect on bone independent of HFD? Or, do the benefits of the OPN neutralizing antibody depend on the insult provided by the HFD? This is essential to understand the impact of HFD-specific macrophage recruitment/osteopontin secretion.

8. Figure 5D. This shows that FC+ cells are increased in eWAT and that this population has increased Spp1 expression. Does this also happen in purified bone marrow FC+ cells after high fat diet? Previous work has shown that osteopontin is produced locally by CD11b+ cells in the bone marrow in mice on high fat diet (PMID 27049717). Figure 6D argues in favor of this since osteopontin is increased in bone marrow supernatant by HFD independent of eWATx (HFD+Sham > NFD+Sham and HFD+eWATx > NFD+eWATx to about the same extent). It is agreed that eWAT also contributes ~40% of the bone marrow supernatant OPN – though this seems to be independent of diet which argues somewhat against the proposed model.

9. The authors acknowledge that macrophage recruitment to the adipose tissue is derived from the bone marrow. However, they then argue that only the macrophages in the gonadal WAT supply osteopontin back to the bone. This conflicts with well-established studies showing that osteopontin is also produced locally by bone marrow macrophages and osteogenic progenitor cells (ex. PMIDs 27049717 and 15845900), including increased production with high-fat diet (PMID 27049717). The focus on eWAT also limits the conclusions, as humans do not have this fat depot. To increase the impact of this work – this reviewer would recommend updating the model to focus more broadly on the peripheral macrophages. For example, if obese humans had mass macrophage recruitment to omental or peri-renal visceral fat – would the same results be expected? Is bone loss proportional to the total macrophage recruitment/systemic burden regardless of where they are localized? Or, is this relationship truly dependent on the mouse-specific eWAT depot?

Minor Concerns:

1. Abstract – Line 25. “macrophages in eWAT are the main source of osteopontin”. Could adjust to “a main source” to allow for other sources of OPN, including macrophages in other fat depots

2. Fig.6 I and J vs Fig.S3. Both say this is anti-human OPN antibody 1- and 24-hours after rOPN-FITC injection. But, they look very different. Please clarify.
3. The authors perform a local injection of neutralizing antibody – however, this seems unlikely to stay local. Were distribution kinetics or serum levels analyzed?
4. Recommend recruiting a statistician to help with the implementation and interpretation of ANOVA results

Response to the editor and reviewers (NCOMMS-20-33071)

Dear editor and reviewers,

Thank you very much for forwarding your recommendation to invite our resubmission. The reviewers' insightful comments are very helpful for improving our work towards publication in your prestigious journal. Based on these comments/suggestions, we have performed additional experiments and addressed the concerns at a point-by-point basis as below. The authors sincerely hope that our replies have adequately addressed your concerns.

Note: Reviewers' comments are highlighted in blue and shown in italic. (Response Fig./Table) means the figure/table used just to answer the questions in our response letter. (Page, Line, Fig.) means the place(s) where the changes are made in the revised manuscript.

REVIEWER #1

In the paper entitled “Macrophages in Epididymal Adipose Tissue Secrete Osteopontin to Regulate Bone Homeostasis” Dai and colleagues showed, that that macrophages in eWAT are the main source of osteopontin, which selectively circulates to the bone marrow and promotes the degradation of bone matrix by activating osteoclasts as well as modulates bone marrow-derived macrophages (BMDMs) to engulf the lipid droplets released from adipocytes within the bone marrow in mice. The Authors showed, that surgical removal of eWAT, local injection of clodronate liposome (for depleting macrophages), and local injection of osteopontin neutralizing antibody show comparable effect on ameliorating HFD-induced bone loss in mice. Presented by the Authors data suggest a future avenue for the development of novel therapeutic strategies to ameliorate obesity-related bone loss. Presented for review Article are well designed and well done. There are major corrections necessary including:

Reply: We appreciate reviewer’s encouragement and constructive comments for improving our work towards publication. In the past 5 months, the authors have conducted additional experiments as advised to improve the study quality and provided the detailed replies together with new data in the specific comments shown below.

COMMENT 1: *Explaining small groups of animals included into the research?*

Reply: Thanks for the question.

To estimate the minimum sample size for both *in vitro* and *in vivo* experiments, we calculated the population mean by our preliminary results (known standard deviation, σ ; margin error, E) and 95% confident ($\alpha = 5\%$, $z_{\alpha/2} = 1.96$) by the formula (Response Fig. 1).

$$n = \left(\frac{z_{\alpha/2} \cdot \sigma}{E} \right)^2$$

Response Fig. 1. The formula for calculation of the minimum sample size (n). E is the maximum error of estimate. $Z_{\alpha/2}$ is the critical value of the normal distribution at $\alpha/2$, and the 95% confidence interval is desired, $Z_{\alpha/2} = 1.96$. σ is the standard deviation.

For example, we calculated the σ (shown in Response Table 2) based on the key μ CT parameter of BV/TV (shown in Response Table 1) in NFD- and HFD-fed mice at week 8. We chose an accuracy of 0.01 for E . The n was acquired as 3 (shown in Response Table 3). The selection of sample size was 5 in our study that attained a statistical significance of $p < 0.01$ with a 95% probability.

Response Table 1. BV/TV in proximal tibiae quantified by μ CT as the key index of bone quality (NFD: normal-fat diet; HFD: high-fat diet.)

NFD					HFD				
0.137	0.1349	0.1559	0.144	0.1443	0.1095	0.0953	0.0905	0.104	0.0953

Response Table 2. The calculated value of σ based on the parameter of BV/TV

NFD	HFD
0.008223	0.007602

Response Table 3. The calculated minimum sample size (n) based on the formula

NFD	HFD
2.597603	2.220076

COMMENT 2: *Explaining why in 12 week there are big secretion of OPN?*

Reply: Thanks for this question. We have included an explanation based on the original result as shown in the revised manuscript (Page 7, Line 132 - 134; Page 11, Line 222 - 224).

Epididymal white adipose tissue (eWAT) secretes osteopontin (OPN) under both physiological (normal-fat diet, NFD) and pathological conditions (high-fat diet, HFD). The amount of secretion depends on the mass of adipose tissue (AT) and the number of infiltrated macrophages.

In response to obesity, previous work shows that eWAT expands dramatically, accompanying with increased infiltration of immune cell, especially macrophages forming crown like structures (CLSs) around dysfunctional adipocytes (1). The percentage of CLSs area in the eWAT of the HFD-fed group was statistically greater than that of the NFD-fed group at week 8, peaking at week 12 (Page 7, Line 132 -134, Fig. 2A). We also found that the concentration of secreted OPN in the HFD/eWAT-derived conditional medium was similar at all time points upon normalized with the tissue mass (Page 11, Line 222 - 224, Fig. 6F). Therefore, the highest value of OPN secretion at week 12 could be explained by the peak mass of eWAT and number of CLSs in HFD-fed mice.

COMMENT 3: *Please rebuild introduction and focus more on OPN and bone density correlation.*

Reply: The constructive comments are well-taken.

We have rebuilt the Introduction in the revised manuscript and also included a paragraph to describe the OPN regulation of bone metabolism as following:

“In bone, OPN mainly regulates the synthesis of extracellular matrix in bone remodeling by stimulating the apposition of osteoclasts to bone matrix (2). OPN is highly concentrated at cement lines where pre-existing and newly formed bone integrate, and OPN-dependent intracellular signaling is seen in this zone for osteoclastic resorption (3). Namely, OPN is involved in organizing the extracellular matrix which coordinates cell-matrix and mineral–matrix interactions. OPN inhibits mineral crystal formation and growth via osteoclasts recruitment and function, on the contrary, OPN-deficient mice possess increased mineral content and crystallinity (4, 5). Up to date, the systemic effects of OPN on the bone homeostasis is not well investigated, especially with regards to eWAT.” (Page 4, Line 64 - 73).

COMMENT 4: *Please in discussion section highlight the potential therapeutic strategy for healing the bone injuries by means of modulation of OPN level.*

Reply: Thanks for the constructive suggestion. We have supplemented the potential therapeutic strategies for bone metabolic disorders in our revised version as:

“Modulation of OPN expression offers the possibility to attenuate versatile bone metabolic disorders based on our present understanding on the roles of OPN. In post-menopausal women with type 2 diabetes, OPN might be a useful marker of bone fracture and worse lipid profile (6). Our results also provide a new insight into the role of eWAT-secreted OPN in the bone homeostasis during obesity, suggesting targeting OPN for effective prevention of bone loss under the condition of HFD feeding. Therefore, applying injectable hydrogels, as our previous investigation(7), combined siRNA *spp1* or OPN neutralizing antibody may promisingly enhance the bone fracture healing via temporo-spatial intervention in obese individuals. Biodegradable membranes or scaffolds that could immobilize OPN neutralizing antibody or deliver OPN receptor antagonist(s), such as integrin antagonists 27, may have broad applications for improving bone regeneration in cases with fracture non-union or bone defects.” (Page 19, Line 411 - 422).

REVIEWER #2

In this study, Dai et al. have explored the relationship between adipose tissue mass and bone homeostasis. Through studies in mice, they have identified osteopontin to be a protein secreted by macrophages in epididymal white adipose tissue that promotes bone matrix degradation. While there are interesting findings presented in the manuscript, important aspects are missing and the resulting picture therefore feels incomplete.

Reply: We appreciate reviewer's insightful comments for improving our work towards publication. We would like to provide the detailed replies together with new data in the specific comments as shown below.

Major points:

COMMENT 1: *As injections with clodronate liposomes have been shown to impact on multiple metabolic processes in mice, e.g. alter insulin sensitivity and protect against diet-induced obesity, additional experiments are required to prove that osteopontin produced by visceral adipose tissue macrophages induce bone remodeling. Could the authors specifically deplete osteopontin from visceral adipose tissue macrophages (as has been done elsewhere, PMID: 23630254) and then measure effects on bone remodeling?*

Reply: Thanks for your constructive comment and advice. We have replenished and discussed these results in our revised version (Page 12, Line 253 - 266, Fig. 7H - L with Supplementary Fig. 11A - D), including Methods (Page 26, Line 552 - 560).

We appreciate the published methods that the reviewer mentioned where an siRNA delivery technology was used to silence inflammatory cytokines expression specifically in eWAT without affecting macrophages in other tissues, including liver, heart, or pancreas. However, this method is inevitable to intervene in the activities of whole phagocytic cells, including macrophages, granulocytes, and dendritic cells.

To explicitly confirm the eWATs-derived OPN involving bone remodeling, we locally injected OPN neutralizing antibody (Neu Ab) into bilateral eWATs of NFD- and HFD-fed mice for 8 weeks. Bone loss was attenuated after injection of Neu Ab in HFD-fed mice, accompanied with a blunting of the ATMs infiltrating in eWAT and decreased accumulation of rBMAT (Response Fig. 2). These data have been included in the revised manuscript (Page 12, Line 253 - 260, Fig. 7H - L and Supplementary Fig. 11A - C).

Besides, we also injected OPN neutralizing antibody (Neu Ab) conjugated with FITC into the unilateral (left side) eWAT of HFD-fed mice. At 1 hour as well as 24 hours post-injection, the FITC signal was situated in locoregional area of injection site rather than other adipose tissues and organs (Response Fig. 3). We have also supplemented these data into the revised version (Page 12, Line 262 - 266, Supplementary Fig. 11D).

These results suggest that the local injection of Neu Ab effectively targets to the eWAT without spreading into the circulatory system and thus exerts little impact on other tissues.

Response Fig. 2 (Fig. 7H-L and Supplementary Fig. 11A-C in the revised manuscript). eWAT-secreted OPN regulates the bone marrow metabolism.

(A) Immunohistochemical staining of MMP9 in the proximal tibiae of mice after injection of saline or Neu Ab into bilateral eWATs for 8 weeks, Scale bar: 100 μ m. (Bottom row is a magnification of the top row, Scale bar: 50 μ m). (B and C) Representative μ CT images (B) and quantification ($n = 5$) (C) of proximal tibiae after mice injected saline or Neu Ab, Scale bar: 500 μ m. (D and E) Representative μ CT images (D) and quantification ($n = 4$) (E) of rBMAT in tibiae after mice injected saline or Neu Ab, Scale bar: 500 μ m. (F) Immunofluorescent staining of macrophages (F4/80) and OPN in eWAT after injection of saline or Neu Ab into bilateral eWATs for 8 weeks.

Red, OPN; green, F4/80; blue, DAPI stain for cell nuclei, Scale bar: 75 μ m. (G and H) Representative images of TRAP staining (Bottom row was magnified from the top row, Scale bar: 50 μ m) (G) and quantification of Trap⁺ osteoclast number and surface ($n = 4$) (H) in the proximal tibiae after mice injected saline or Neu Ab, Scale bar: 100 μ m. Oc.N/BS, osteoclast number per bone surface; Oc.S/BS, osteoclast surface per bone surface. All data are presented as mean \pm SD. * $P < 0.05$; ** $P < 0.01$; *** $P < 0.001$; by two-way ANOVA with *Tukey's post hoc* test (C, E, and H).

Response Fig. 3 (Supplementary Fig. 11D in the revised Supplementary File). *Ex vivo* imaging (IVIS200 system) of the indicated organs and tissues.

Ex vivo imaging of the indicated organs and tissues collected at 1 hour and 24 hours after unilateral injection of Neu Ab conjugated with FITC (Neu Ab-FITC) into the eWAT.

COMMENT 2: *There is no difference in circulating osteopontin levels comparing mice fed a normal or high-fat diet. Still, the authors claim that increased expression of osteopontin during weight gain allows crosstalk between fat and bone. It is suggested that osteopontin receptor levels are key to this, but there is no difference comparing immunohistochemical staining of OPN bound to cells in proximal tibiae of NFD and HFD-fed groups. Can the authors provide evidence on exactly how this works?*

Reply: Thanks for the question.

To address this question, we would like to specify the difference in the statement “OPN-positive stained cells” and “OPN-positive stained cells/BS (bone surface)” as provided in Response Fig. 4. OPN-positive stained cells include Part 2 and 3. OPN-positive stained cells/BS (bone surface) refers to Part 3. The expression of OPN receptor ($\alpha\beta 3$) in bone marrow is mainly derived from the Part 2.

Response Fig. 4. The definition of versatile OPN in bone marrow. OPN-positive stained cells include Part 2 and 3. OPN-positive stained cells/BS (bone surface) refers to Part 3. The expression of OPN receptor ($\alpha\nu\beta 3$) in bone marrow is mainly derived from the Part 2.

We showed here similar concentrations of free OPN (Part 1) as well as OPN-positive stained cells (Part 2 and 3, Page 10, Line 204 - 207, Fig. 6A, B) in proximal tibia of NFD- and HFD-fed mice. Besides, the expression of OPN receptor ($\alpha\nu\beta 3$) in bone marrow of HFD-fed mice were much higher than that in NFD-fed mice at week 12 (Page 12, Line 251 - 253, Fig. 7G), while the expression was significantly higher in BMDM than BMSC (Page 12, Line 243 - 246, Fig. 7A), indicating a possibility of elevated biological activities from BMDM.

Subsequently, we found that the percentage of FC⁺ (F4/80⁺, CD11b⁺) and OPN⁺ BMDMs in HFD-fed group was higher than that in NFD-fed group at weeks 8 and 12 (Response Fig. 5). Also, the proportion of FC⁺ and perilipin 1⁺ BMDMs was statistically higher in the HFD-fed group than that in the NFD-fed group at weeks 8 and 12 (Page 14 - 15, Line 309 - 313, Fig. 9F, G). These results validate a higher proportion of executive OPN in HFD-fed mice that echoes the increased number OPN receptors in bone marrow.

Response Fig. 5. Percentage of FC+ and OPN+ BMDMs.

FACS analysis of FC+ (F4/80+ and CD11b+, top) and OPN+ (bottom) BMDMs from NFD- (left) and HFD-fed (right) mice. P9, FC+ and OPN+ BMDM population from Q2 (P8). Percentage of FC+ and OPN+ BMDMs at weeks 8, 12, and 16 (right, $n = 5$). All data are presented as mean \pm SD. ** $P < 0.01$; *** $P < 0.001$; by two-way ANOVA with *Sidak's post hoc* test.

REVIEWER #3

The authors present an interesting study that reaffirms and extends the notion that macrophage recruitment to visceral white adipose tissue depots contributes to the negative effects of visceral obesity on bone. They show that this effect is mediated primarily by the release of circulating osteopontin by the macrophages within the gonadal fat depot, subsequently inducing a cascade of events within the bone and bone marrow that impact osteoclast function and osteoclastogenesis. This manuscript builds upon their previous work and other studies in the field showing that osteopontin is a critical mediator of high-fat diet induced adverse phenotypes. Currently, there are key issues with the data that need to be considered. However, if addressed, this study would be an excellent addition to the literature.

Reply: Many thanks for recognizing the significance of this study. We have conducted additional experiments to address your concerns, with details in the replies below.

Major Concerns:

COMMENT 1: *Figure 1. Why does the eWAT and rBMAT recover at 16-weeks? Did you happen to have a group of non-responders (selection bias) or is this recovery despite ongoing HFD consumption a repeatable result?*

Reply: We appreciate the reviewer's concerns and suggestion to validate the proposed model.

We confirmed the 'recuperative' masses of rBMAT and eWAT in ongoing NFD/HFD consumption by two additional repeats. The phenomena are consistently observed in the HFD-fed mice, not a selection bias. In the revised manuscript, we have discussed the published work from others, which showed the similar phenomena mentioned above (Page 3, Line 52 - 56).

Several available experimental studies also report that eWAT mass strongly expands at week 6 and peaks at week 12 of HFD-fed mice (8, 9). Highlights outline that hypertrophy of adipocytes is associated with an increase in a hypoxic experience in visceral adipose tissue rather than subcutaneous adipose tissue. Hypoxic adipocytes often undergo necrosis, leading to infiltration of immune cells and tissue inflammation (10-12) (Page 3, Line 52 - 56). Infiltrated ATMs clear the dead adipocytes by the lysosomal compartments, resulting in the formation of CLSs. The transferred lipid is reported to be accumulated in the liver and other tissues (13).

We have validated the consistent tendency of rBMAT and eWAT mass in response to diet and found out certain cytokine cascading of the bone disorders in rBMAT. We have also addressed why the mass of rBMAT recovered after HFD feeding for 12 weeks based on findings of our current experiments (Page 5, Line 99 - 103).

COMMENT 2: *Figure 2. What about other visceral WAT depots? In rodents, the perirenal WAT is a large visceral depot that more closely replicates visceral WAT in humans (humans do not have an eWAT depot). There are also adipocytes present around the omental tissues.*

Reply: Suggestion is well-taken. We have replenished the additional results of other VATs in our revised version (Page 7, Line 143 - 146; Page 9, Line 183 - 186; Page 10 - 11, Line 217 - 219, with Supplementary Fig. 2A - D; Supplementary Fig. 4A, B; and Supplementary Fig. 9A, B), including Methods (Page 27, Line 575 - 581).

One distinction between human and rodent visceral white adipose tissue (VAT) is that humans have detectable omental WAT while the omental WAT is often only visible in obese rodents). Instead, rodents have large eWAT (14, 15). Therefore, we complementally analyzed the alterations of inflammatory responses in perirenal white adipose tissue (PAT) and mesenteric white adipose tissue (MAT) for our current work.

Notably, the percentage of CLS area in PAT was similar between the two groups at all time points, except for week 16. Besides, the percentage of CLS area in the MAT of the HFD-fed group was statistically lower than that of the NFD-fed group at weeks 4 and 12 (Response Fig. 6A and 6B, in Page 7, Line 143 - 146, Supplementary Fig. 2A, B). HFD regulation had no discernable effects on the gene expression of *Tnfa*, *Il-1b*, and *Il-10* in PAT over the course of the feeding regimens but had a slightly elevated gene expression in MAT at week 12 of NFD-fed mice, indicating less inflammatory responses in PAT and MAT of HFD-fed mice (Response Fig. 6C and 6D, in Page 7, Line 143 - 146, Supplementary Fig. 2C, D).

We subsequently found that the expression of *Spp1* was comparable between NFD- and HFD-fed groups no matter in PAT or MAT, except for the week 16 in PAT, which was higher in the HFD-fed group as compared with the NFD-fed group (Response Fig. 7A and 7B, in Page 9, Line 183 - 186, Supplementary Fig. 4A, B). Besides, we surgically removed the bilateral PATs (PATx) at 12-week-old mice before the initiation of the indicated diets. The concentrations of OPN in serum of the PATx groups were similar as compared with that in the Sham groups at weeks 8 and 12 (Response Fig. 8A and 8B, in Page 10 - 11, Line 217 - 219, Supplementary Fig. 9A, B), indicating the small proportion of PAT-derived OPN existing in the circulatory system. These results demonstrate that PAT and MAT does not induce the metabolic disorders of bone marrow in HFD-fed mice.

Response Fig. 6 (Supplementary Fig. 2 in the revised Supplementary File). The inflammation in PAT and MAT.

(A) Representative H&E staining (left) and quantification of CLS area percentage (right, $n = 4$) of PAT from NFD- and HFD-fed mice at the indicated time points, Scale bar: 100 μm . (B) Representative H&E staining (left) and quantification of CLS area percentage (right, $n = 3$) of MAT from NFD- and HFD-fed mice at the indicated time points. Scale bar, 100 μm . (C and D) Relative expression of *Tnfa*, *Il-1b*, and *Il-10* in PAT ($n = 4$) (C) and MAT ($n = 4$) (D), respectively, over the course of the feeding regimens. All data are presented as mean \pm SD. * $P < 0.05$; ** $P < 0.01$; *** $P < 0.001$; by two-way ANOVA with *Sidak's post hoc* test (A, B, C, and D).

Response Fig. 7 (Supplementary Fig. 4 in the revised Supplementary File). Relative expression of *Spp1* in PAT and MAT.

(A and B) Relative expression of *Spp1* in PAT (A) and MAT (B) over the course of the feeding regimens ($n = 4$). All data are presented as mean \pm SD. * $P < 0.05$; ** $P < 0.01$; *** $P < 0.001$; by two-way ANOVA with *Sidak's post hoc* test (A and B).

Response Fig. 8 (Supplementary Fig. 9 in the revised Supplementary File). OPN expression in serum.

(A and B) OPN expression in serum of NFD- and HFD-fed groups with or without PATx at weeks 8 (A) and 12 (B), ($n = 5$). All data are presented as mean \pm SD, by two-way ANOVA with *Tukey's post hoc* test (A and B).

COMMENT 3: *Figure 3. Does eWAT removal actually rescue bone? The current results are relatively inconclusive and underpowered. At 8-weeks the answer is clearly 'no'. At 12-weeks there is a slight improvement in BV/TV, but it does not appear that the 'eWATx' x HFD interaction effect would be significant by ANOVA. As this is the core observation for rest of the paper – it should be strengthened/repeated to confirm or deny the result. The authors could also consider removing the other visceral fat (ex. perirenal fat) if this is found to be a source of osteopontin.*

Reply: Thank you for the constructive comments. We have confirmed that eWAT removal rescued bone loss by conducting additional experiment. We also supplemented the results of surgical removal of bilateral PATs in the revised manuscript as you suggested (Page 8, Line 154 - 155; Page 10 - 11, Line 217 - 219 with Supplementary Fig. 3A, B and Supplementary Fig. 9A, B).

We found that the trabecular bone mass in the HFD + eWATx group was

significantly greater than that of the HFD + Sham group at week 12 post-surgery instead of week 8 (This description has been presented in the original manuscript and now also in the revised version on Page 7 - 8, Line 149 - 152). The bone mass in HFD + eWATx group was found above 30% higher than that in HFD + Sham group, which almost approximated the mass in NFD + Sham group. Besides, we combined other specific methods to confirm the reciprocal relationship between eWAT-derived OPN and bone marrow, e.g. local injection of clodronate liposomes for depleting ATMs and local injection of Neu Ab.

We surgically removed the bilateral PATs (PATx) at 12-week-old mice before the initiation of the indicated diets. We found that the trabecular bone mass in the PATx groups were similar to that in the Sham groups at weeks 8 and 12 post-surgery (Response Fig. 9A and 9B, in Page 8, Line 154 - 155, Supplementary Fig. 3A, B). Besides, the concentration of OPN in serum of the PATx groups were similar as compared with that in the Sham groups at weeks 8 and 12 (Response Fig. 9C, in Page 10 - 11, Line 217 - 219, Supplementary Fig. 9A, B). These results demonstrate that PAT or PAT-derived OPN might not be the cause to induce the metabolic disorders of bone marrow in HFD-fed mice.

Response Fig. 9 (Supplementary Fig. 3 and Supplementary Fig. 9 in the revised Supplementary File). The bone mass of NFD- and HFD-fed mice with or without removal of bilateral PATs (PATx).

(A and B) Representative μ CT images (A) and quantification ($n = 5$) (B) of proximal tibiae from NFD- and HFD-fed mice that had undergone either sham surgery or removal of bilateral PATs (PATx), Scale bar: 500 μ m. (C) OPN expression in serum of NFD- and HFD-fed groups with or without PATx at weeks 8 and 12 ($n = 5$). All data are presented as mean \pm SD. * $P < 0.05$; ** $P < 0.01$; *** $P < 0.001$; by two-way ANOVA with *Tukey's post hoc* test (B and C).

COMMENT 4: *Figure 4. Missing NFD + clodronate liposome control group. Do clodronate liposomes impact bone in the NFD group, independent of HFD? Please show individual data points (as in Fig.3).*

Reply: Thanks for the comments and suggestions.

We have performed additional experiments to supplement the NFD + Clodronate Liposome (NFD+CL) group (Response Fig. 10, Page 8, Line 166 - 171, Fig. 4A - E). We have shown individual data points in these groups as you suggested.

Response Fig. 10 (Fig. 4A - E in the revised manuscript). Infiltrating ATMs in eWAT are related to bone metabolism.

(A) Immunofluorescent staining of macrophage (F4/80) in eWAT after mice injected with clodronate liposome (CL) or control liposome (Ctrl) into bilateral eWATs for 8 weeks. Green, F4/80; blue, DAPI stain for cell nuclei; Scale bar: 75 μ m. (B) Quantification of F4/80-positive area percentage for per unit area in immunofluorescent staining from A ($n = 5$). (C and D) Representative μ CT image (C) and quantification ($n = 5$) (D) of proximal tibiae after mice injected with control liposome or CL for 8 weeks. (E) μ CT quantification of rBMAT in tibiae after mice injected with control liposome or CL for 8 weeks ($n = 4$). All data are presented as mean \pm SD. * $P < 0.05$; ** $P < 0.01$; *** $P < 0.001$; by two-way ANOVA with *Tukey's post hoc* test (B, D, and E).

COMMENT 5 : *Figure 5. The presented immunostaining results look like autofluorescence of the entire adipocyte extracellular matrix. Additional validation of the antibody with positive and negative controls is needed. It is agreed that given the large increase in mRNA, corresponding increases in protein are likely.*

Reply: Your constructive comments are well taken.

We have replenished the immunofluorescent staining with appropriate ones and added the additional validation of versatile controls as you suggested (Response Fig. 11A and 11B, Page 9, Line 188 - 191, Fig. 5A and supplementary Fig. 5). Immunohistochemical staining of OPN and F4/80 in serial sections of eWAT also echo the similar situs of immunofluorescent staining, eliminating bias of potential autofluorescence (Response Fig. 11C).

Response Fig. 11 (Fig. 5A and Supplementary Fig. 5 in the revised manuscript and Supplementary File). Representative images of immunofluorescent and immunohistochemical staining of OPN and macrophage in eWAT.

(A) Representative images of immunofluorescent staining of OPN and macrophage (F4/80) in eWAT from NFD- and HFD-fed mice at weeks 8, 12, and 16, Scale bar: 100 μm . (B) Control immunofluorescent staining of the OPN and F4/80 in eWAT, Scale bar: 100 μm . (C) Immunohistochemical staining of OPN and F4/80 in serial sections of eWAT.

COMMENT 6: *Figure 6. Same comments about OPN IHC. The bone images provided are not suitable to localize the staining and appropriate positive and negative controls are needed. In addition, what happens to Spp1 expression in the floating fraction in 6B? In addition to adipocytes, this fraction would presumably contain any lipid-laden macrophages.*

Reply: Thanks for the comments and suggestions.

We have replaced the mentioned images with appropriate ones and added the additional validation of versatile controls as you suggested (Response Fig. 12, in Page 10, Line 204 - 207, Fig. 6A and Supplementary Fig. 6).

Due to their large size and buoyance, those lipid-laden macrophages cannot be captured by pelleting and cell sorting, and furthermore, they are rare populations in the bones as reported before (16). Therefore, this also explains why we failed to acquire the *spp1* expression in the floating fraction of bone marrow.

However, our original results of immunofluorescent staining in a blood smear showed that lipid-laden F4/80+ BMDMs were present in buoyant samples from the bone marrow supernatant of HFD-fed group as indicated by expression of perilipin 1, a marker of lipid droplets (Response Fig. 13A and 13B, in Page 14, Line 306 - 309, Fig. 6B and 9E). We sorted the BMDMs from the centrifuged pellets by binning FC+ macrophages into perilipin 1+ group and perilipin 1- group. The proportion of FC+ and perilipin 1+ BMDMs was statistically greater in the HFD-fed group than that in the NFD-fed group at weeks 8 and 12 (Response Fig. 13C and 13D, in Page 14 - 15, Line 309 - 312, Fig. 9F, G). This relative relationship between two groups was similar to that of free fatty acid (FFA) in bone marrow (Page 15, Line 312 - 313, Supplementary Fig. 13). These original results together suggest the elevated lipo-phagocytosis of BMDM in HFD-fed mice.

Response Fig. 12 (Fig. 6A and Supplementary Fig. 6 in the revised manuscript and Supplementary File). Representative images of immunohistochemical staining OPN.

(A) Representative images of immunohistochemical staining OPN in proximal tibiae of NFD- and HFD-fed groups at weeks 4, 8, 12, and 16, Scale bar: 50 μ m. (B) Control immunohistochemical staining of OPN in proximal tibiae, Scale bar: 50 μ m.

Response Fig. 13 (Fig. 6B and Fig. 9E-G in the revised manuscript). The lipid-

laden macrophages in bone marrow.

(A) Schematic representation of isolation of bone marrow. (B) Immunofluorescent images of buoyant BMDM (F4/80), with or without containing lipid vesicles (Perilipin 1), isolated from bone marrow supernatant at week 12, Scale bar: 15 μ m. (C) FACS analysis of FC⁺ (F4/80⁺ and CD11b⁺, top) and Perilipin1⁺ (bottom) BMDMs from the bone marrow pellet of NFD- (left) and HFD-fed (right) mice. P9, FC⁺ and Perilipin1⁺ BMDM population from Q2. (D) Percentage of FC⁺ and Perilipin1⁺ BMDMs at weeks 8, 12, and 16 ($n = 5$). All data are presented as mean \pm SD. *** $P < 0.001$; by two-way ANOVA with *Sidak's post hoc* test (D).

COMMENT 7: *Figure 7. Missing control diet + saline and control diet + Neu Ab control groups. Does the OPN neutralizing antibody have the same positive effect on bone independent of HFD? Or, do the benefits of the OPN neutralizing antibody depend on the insult provided by the HFD? This is essential to understand the impact of HFD-specific macrophage recruitment/osteopontin secretion.*

Reply: Thanks for this insightful comment. We have conducted additional experiments to supplement the control diet + saline and control diet + Neu Ab groups in Fig. 7H - L with Supplementary Fig. 11A - D and Methods (Page 26, line 552 - 560).

Bone loss was attenuated after injection of Neu Ab in HFD-fed mice, accompanied with a blunting of the ATMs infiltrating in eWAT and decreased accumulation of rBMAT (Response Fig. 14, in Fig. 7H - L and Supplementary Fig. 11A - C). This updated result shows that Neu Ab does not increase bone mass in the mice fed with control diet (NFD). That is, the benefits of the OPN Neu Ab depend on the insult provided by the HFD (Page 12, line 260 - 262).

Response Fig. 14 (Fig. 7H-7L and Supplementary Fig. 11A-C in the revised manuscript and Supplementary File). eWAT-secreted OPN regulates the bone marrow metabolism.

(A) Immunohistochemical staining of MMP9 in the proximal tibiae of mice after injection of saline or Neu Ab into bilateral eWATs for 8 weeks, Scale bar: 100 μm. (Bottom row is a magnification of the top row, Scale bar: 50 μm). (B and C) Representative μCT images (B) and quantification ($n = 5$) (C) of proximal tibiae after mice injected saline or Neu Ab, Scale bar: 500 μm. (D and E) Representative μCT images (D) and quantification ($n = 4$) (E) of rBMAT in tibiae after mice injected saline or Neu Ab, Scale bar: 500 μm. (F) Immunofluorescent staining of macrophages (F4/80) and OPN in eWAT after injection of saline or Neu Ab. Red, OPN; green, F4/80; blue, DAPI stain for cell nuclei, Scale bar: 75 μm. (G and H) Representative images of TRAP staining (Bottom row is a magnification of the top row, Scale bar: 50 μm) (G) and quantification of Trap+ osteoclast number and surface ($n = 4$) (H) in the proximal tibiae

after mice injected saline or Neu Ab, Scale bar: 100 μ m. Oc.N/BS, osteoclast number per bone surface; Oc.S/BS, osteoclast surface per bone surface. All data are presented as mean \pm SD. * $P < 0.05$; ** $P < 0.01$; *** $P < 0.001$; by two-way ANOVA with Tukey's *post hoc* test (C, E, and H).

COMMENT 8: *Figure 5D. This shows that FC+ cells are increased in eWAT and that this population has increased Spp1 expression. Does this also happen in purified bone marrow FC+ cells after high fat diet? Previous work has shown that osteopontin is produced locally by CD11b+ cells in the bone marrow in mice on high fat diet (PMID 27049717). Figure 6D argues in favor of this since osteopontin is increased in bone marrow supernatant by HFD independent of eWATx (HFD+Sham > NFD+Sham and HFD+eWATx > NFD+eWATx to about the same extent). It is agreed that eWAT also contributes ~40% of the bone marrow supernatant OPN – though this seems to be independent of diet which argues somewhat against the proposed model.*

Reply: Thanks for the comments and suggestions.

We have replenished the *Spp1* expression in FC+ cells from bone marrow in the revised manuscript as you suggested (Page 10, Line 207 - 210, Supplementary Fig. 7A - C).

Our study showed that FC+ BMDMs were gated by FACS from bone marrow after NFD and HFD feeding for 12 weeks. Although the mRNA expression of *Spp1* was similar in both FC+ BMDMs with or without HFD feeding, the proportion of FC+ BMDMs was significantly lower in the HFD-fed group as compared with the NFD-fed group at week 12 post feeding (Response Fig. 15A - 15C, in Page 10, Line 207 - 210, Supplementary Fig. 7A - C). These results convey the message that while OPN can be locally produced by FC+ cells in bone marrow, the number of FC+ BMDMs in HFD-fed mice was arrested as compared to that in NFD-fed mice.

Response Fig. 15 (Supplementary Fig. 7 in the revised Supplementary File). The relative expression of *spp1* in FC+ BMDMs.

(A) Macrophages gated by FACS for F4/80 and CD11b expression from bone marrow at week 12 of NFD (left) and HFD (right) feeding. Q2 represents the F4/80+ and

CD11b+ (FC+) macrophages population. (B) The percentage of FC+ BMDMs from NFD- and HFD-fed mice at week 12 ($n = 4$). (C) Relative expression of *Spp1* in FC+ BMDMs from NFD- and HFD-fed mice at week 12 ($n = 4$). All data are presented as mean \pm SD. *** $P < 0.001$; by two-tailed *Welch's t*-test (B and C).

COMMENT 9: *The authors acknowledge that macrophage recruitment to the adipose tissue is derived from the bone marrow. However, they then argue that only the macrophages in the gonadal WAT supply osteopontin back to the bone. This conflicts with well-established studies showing that osteopontin is also produced locally by bone marrow macrophages and osteogenic progenitor cells (ex. PMIDs 27049717 and 15845900), including increased production with high-fat diet (PMID 27049717). The focus on eWAT also limits the conclusions, as humans do not have this fat depot. To increase the impact of this work – this reviewer would recommend updating the model to focus more broadly on the peripheral macrophages. For example, if obese humans had mass macrophage recruitment to omental or peri-renal visceral fat – would the same results be expected? Is bone loss proportional to the total macrophage recruitment/systemic burden regardless of where they are localized? Or, is this relationship truly dependent on the mouse-specific eWAT depot?*

Reply: Thanks for the insightful comment. The original results (Fig. 6A - E) and additional experiments (Supplementary Fig. 7 in the revised Supplementary File) could provide an answer for addressing your concern. In fact, OPN is also produced locally by bone marrow macrophages and osteogenic progenitor cells. These results also point to the possibility that the circulating OPN might be an endogenous supplement for the scarcity in bone marrow (Page 11, Line 238 - 240). Besides, we have incorporated these important comments below into the Result and Discussion sections of the revised manuscript (Page 14 - 15, Line 306 - 313; Page 19, Line 423 - 425).

We appreciate the profound influence of eWAT secreted OPN on bone homeostasis, and our closer view on the intrinsic relationship has not excluded other possibilities. Our current study neither eliminates the possibility of OPN from other sources of deposits (for example, liver and kidney) nor excludes other cytokines (for example, $IL-1\beta$ and $TNF-\alpha$) involved in bone marrow metabolism. We have specified this limitation in our revised version (Page 19, Line 423 - 425).

In rodent models, the iWAT (around the groin) and eWAT (or gonadal WAT) are the predominant models of subcutaneous and visceral adipose tissue, respectively (10, 17-20). eWAT is the primary storage depot that grows initially in HFD-induced obesity, followed by subcutaneous AT and mesenteric AT (21). eWAT is originally considered as an organ with homogeneous function, but it is now thought to exert more adverse effects on health compared with subcutaneous adipose tissue (17). Indeed, the stromal

vascular fraction of the eWAT contains a higher fraction of macrophages compared with iWAT, MAT, and other ATs. Also, the absolute number of macrophages per AT depot is higher in the eWAT than the other AT depots (Fig. 2A, B; Supplementary Fig. 2A, B).

Our original results are consistent with a recent report showing that HFD-induced obesity leads to a dramatic increase in the number of lipid-associated macrophages (LAM) at the expense of non-perivascular-like macrophages and perivascular-like macrophages in eWAT, rather than other VATs (Page 14 - 15, Line 306 - 313, Fig. 9E - G and Supplementary Fig. 13) (11). Besides, *osteopontin* (also known as *Spp1*) is the most highly upregulated cytokines in ATMs in obesity as we shown (11). Subsequently, we corroborate that the surgical removal of PATs has no discernable effects on trabecular bone mass and OPN concentration in serum, partly explaining the HFD-induced bone loss truly depends on the mouse-specific eWAT depot (Fig. 3, Supplementary Fig. 3 and Supplementary Fig. 9), although our results could not fully eliminate the possibility of other ATs-derived insult on bone homeostasis.

Humans have detectable omental AT (the omental AT is often only visible in obese rodents), whereas rodents have large eWAT that is a rare portion in human (14, 15). However, these divergences do not impede the conceptual application in human. Single-cell analysis of human omental adipose tissue identifies that obesity activates a non-classic inflammatory phenotype of ATM involving lipid accumulation and trafficking (22). Moreover, plasma OPN and mRNA expression of *spp1* in omental adipose tissue are higher in obese individuals than lean individuals (23). Nevertheless, the certain linking correlation by OPN between VAT and bone metabolism in human remains elusive.

Minor Concerns,

COMMENT 10: *Abstract – Line 25. “macrophages in eWAT are the main source of osteopontin”. Could adjust to “a main source” to allow for other sources of OPN, including macrophages in other fat depots.*

Reply: Suggestion is well-taken.

We have changed “macrophages in eWAT are the main source of osteopontin” to “macrophages in eWAT are a main source of osteopontin” in the revised manuscript (Page 2, Line 26 - 27).

COMMENT 11: *Fig.6 I and J vs Fig.S3. Both say this is anti-human OPN antibody 1- and 24-hours after rOPN-FITC injection. But, they look very different. Please clarify.*

Reply: Thanks for the careful review.

Endogenous OPN expression secreted by mouse was negatively stained by anti-human OPN antibody in these organs and tissues. Accordingly, we used the anti-human

OPN antibody to evaluate the temporo-spatial domains of exogenous recombination human-OPN in our study where the mice were injected with the rOPN-FITC (Page 11, Line 229 - 236, Fig. 6I, J). We eliminated the biological reactions of endogenous mouse OPN with anti-human OPN antibody (Page 11, Line 237 - 238, Supplementary Fig. 10). Therefore, these differences suggest that positive stained area in our revised Fig. 6J can be obviously less than that in the revised Fig. 6I due to the specific targets in bone and eWAT, and OPN IHC staining shown in the revised Fig. S10 is negatively stained due to difference in animal species.

COMMENT 12: *The authors perform a local injection of neutralizing antibody – however, this seems unlikely to stay local. Were distribution kinetics or serum levels analyzed?*

Reply: Thanks for the comments.

We have performed additional experiments to offer the reliable results in the revised manuscript as you suggested (Page 12 - 13, Line 262 - 266, Supplementary Fig. 11D).

We injected Neu Ab conjugated with FITC into the unilateral (left side) eWAT of HFD-fed mice. At 1 hour as well as 24 hours post-injection, the FITC signal was situated in locoregional area of injection site rather than other AT and organs (Response Fig. 16, in Page 12 - 13, Line 262 - 266, Supplementary Fig. 11D). These results suggest that the injection of Neu Ab into unilateral eWAT is still in the localized position.

Besides, previous study reported that mouse received 50 μg OPN neutralizing antibody that was much more than 1.25 μg /unilateral/3 days used in our study (24). Therefore, we may confirm that even the local injection of OPN neutralizing antibody influences the systemic metabolism, and such insignificant amount might have less confounding effect.

Response Fig. 16 (Supplementary Fig. 11D in the revised Supplementary File). **Ex vivo** imaging of the indicated organs and tissues after unilateral injection of Neu Ab-FITC into the eWAT.

Ex vivo imaging (IVIS200 system) of the indicated organs and tissues collected at 1 hour (left part in the right box) and 24 hours (right part in the right box) after unilateral injection of OPN neutralizing antibody (Neu Ab-FITC) into the eWAT.

COMMENT 13: *Recommend recruiting a statistician to help with the implementation and interpretation of ANOVA results.*

Reply: Suggestion is well-taken. We have asked a biostatistician to check the statistical analysis throughout this study. We confirm that all the data have been appropriately presented and interpreted to the best of our knowledge.

References

1. Q. A. Wang, C. Tao, R. K. Gupta, P. E. Scherer, Tracking adipogenesis during white adipose tissue development, expansion and regeneration. *Nature medicine* **19**, 1338-1344 (2013).
2. S. R. Rittling *et al.*, Mice lacking osteopontin show normal development and bone structure but display altered osteoclast formation in vitro. *Journal of bone and mineral research : the official journal of the American Society for Bone and Mineral Research* **13**, 1101-1111 (1998).
3. M. D. McKee, A. Nanci, Osteopontin and the bone remodeling sequence. Colloidal-gold immunocytochemistry of an interfacial extracellular matrix protein. *Ann N Y Acad Sci* **760**, 177-189 (1995).
4. C. L. Duvall, W. R. Taylor, D. Weiss, A. M. Wojtowicz, R. E. Guldberg, Impaired angiogenesis, early callus formation, and late stage remodeling in fracture healing of osteopontin-deficient mice. *Journal of bone and mineral research : the official journal of the American Society for Bone and Mineral Research* **22**, 286-297 (2007).
5. A. L. Boskey, L. Spevak, E. Paschalis, S. B. Doty, M. D. McKee, Osteopontin deficiency increases mineral content and mineral crystallinity in mouse bone. *Calcified tissue international* **71**, 145-154 (2002).
6. T. Filardi *et al.*, High serum osteopontin levels are associated with prevalent fractures and worse lipid profile in post-menopausal women with type 2 diabetes. *Journal of endocrinological investigation* **42**, 295-301 (2019).
7. Q. Feng *et al.*, Dynamic and Cell-Infiltratable Hydrogels as Injectable Carrier of Therapeutic Cells and Drugs for Treating Challenging Bone Defects. *ACS central science* **5**, 440-450 (2019).
8. P. Mulder *et al.*, Surgical removal of inflamed epididymal white adipose tissue attenuates the development of non-alcoholic steatohepatitis in obesity. *International journal of obesity (2005)* **40**, 675-684 (2016).
9. K. J. Strissel *et al.*, Adipocyte death, adipose tissue remodeling, and obesity complications. *Diabetes* **56**, 2910-2918 (2007).
10. A. L. Ghaben, P. E. Scherer, Adipogenesis and metabolic health. *Nature reviews*.

- Molecular cell biology* **20**, 242-258 (2019).
11. A. K. Sárvári *et al.*, Plasticity of Epididymal Adipose Tissue in Response to Diet-Induced Obesity at Single-Nucleus Resolution. *Cell metabolism* **33**, 437-453.e435 (2021).
 12. D. A. Jaitin *et al.*, Lipid-Associated Macrophages Control Metabolic Homeostasis in a Trem2-Dependent Manner. *Cell* **178**, 686-698.e614 (2019).
 13. B. R. Coats *et al.*, Metabolically Activated Adipose Tissue Macrophages Perform Detrimental and Beneficial Functions during Diet-Induced Obesity. *Cell reports* **20**, 3149-3161 (2017).
 14. Q. Luong, J. Huang, K. Y. Lee, Deciphering White Adipose Tissue Heterogeneity. *Biology* **8**, (2019).
 15. D. P. Bagchi, O. A. MacDougald, Identification and Dissection of Diverse Mouse Adipose Depots. *Journal of visualized experiments : JoVE*, (2019).
 16. L. Zhong *et al.*, Single cell transcriptomics identifies a unique adipose lineage cell population that regulates bone marrow environment. *eLife* **9**, (2020).
 17. L. van Beek *et al.*, The limited storage capacity of gonadal adipose tissue directs the development of metabolic disorders in male C57Bl/6J mice. *Diabetologia* **58**, 1601-1609 (2015).
 18. S. Nishimoto *et al.*, Obesity-induced DNA released from adipocytes stimulates chronic adipose tissue inflammation and insulin resistance. *Science advances* **2**, e1501332 (2016).
 19. S. E. Flaherty, 3rd *et al.*, A lipase-independent pathway of lipid release and immune modulation by adipocytes. *Science (New York, N.Y.)* **363**, 989-993 (2019).
 20. X. Xu *et al.*, Obesity activates a program of lysosomal-dependent lipid metabolism in adipose tissue macrophages independently of classic activation. *Cell metabolism* **18**, 816-830 (2013).
 21. L. Sackmann-Sala, D. E. Berryman, R. D. Munn, E. R. Lubbers, J. J. Kopchick, Heterogeneity among white adipose tissue depots in male C57BL/6J mice. *Obesity (Silver Spring, Md.)* **20**, 101-111 (2012).
 22. J. Vijay *et al.*, Single-cell analysis of human adipose tissue identifies depot and disease specific cell types. *Nature metabolism* **2**, 97-109 (2020).
 23. J. Gómez-Ambrosi *et al.*, Plasma osteopontin levels and expression in adipose tissue are increased in obesity. *The Journal of clinical endocrinology and metabolism* **92**, 3719-3727 (2007).
 24. Y. Hirano *et al.*, Neutralization of osteopontin attenuates neutrophil migration in sepsis-induced acute lung injury. *Critical care (London, England)* **19**, 53 (2015).

REVIEWERS' COMMENTS

Reviewer #2 (Remarks to the Author):

The authors have addressed all questions.

Reviewer #3 (Remarks to the Author):

The authors have thoroughly addressed my concerns. Two minor comments are recommended to improve the impact of the final manuscript.

1. 2-way ANOVA results should be reported throughout to promote accurate interpretation of the data. For example, all panels of Fig.3B should be analyzed by 2-way ANOVA (Diet x eWATx) to determine the p values for the main and interaction effects for diet type and eWATx. 2-way ANOVA P-values for main and interaction effects should be reported in the figure legends or as a supplemental table(s) for relevant graphs. This includes Fig.3B,D; Fig4D,E; Fig.6D; Fig.7I,J,L; etc.
2. Fig.6A. Please add the units to the graph.

Reviewer #4 (Remarks to the Author):

Response to Reviewer #1 rebuttal

Comment 1 (explain the small groups of animals used) - The authors have adequately explained the requested sample size calculations.

Comment 2 (why is there high OPN at 12 weeks) - The authors have explained clearly why large amounts of OPN occur at 12 weeks. This correlates well, as expected, with high macrophage numbers in the crown like structures (CLSs) surrounding the eWAT of the HFD mouse group.

Comment 3 (revise Intro to focus more on OPN and bone density) - There are some major issues with the added paragraph, and I have corrected as follows:

"OPN secreted predominantly by osteoblasts and osteocytes is an abundant noncollagenous phosphoprotein of the extracellular matrix of bone. There it serves to regulate matrix mineralization and bone cell (osteocyte and osteoclast) attachment (16-19). OPN is highly concentrated at cement lines where pre-existing and newly formed bone integrate (mineral-matrix interactions), and at bone surfaces in the so-called lamina limitans structure surrounding osteocytes and underlying osteoclasts (cell-matrix interactions) (17). OPN inhibits mineralization by stabilizing mineral precursor phases and/or by binding to crystal surfaces (doi: 10.1016/j.bone.2020.115447), and OPN-deficient mice possess increased mineral content and crystallinity (4,5). OPN-dependent intracellular signaling is seen during initial osteoclast attachment to bone surfaces and during the osteoclastic resorption phase (3), and thus OPN appears to be an important player in the bone remodeling cycle. To date, the systemic effects of OPN on bone homeostasis have not been well studied, especially with regard to the biology of eWAT. Here we further investigate links between bone, OPN and eWAT."

Comment 4 (highlight in Discussion potential therapeutic strategies for modulating OPN for bone healing). There are some issues with the added paragraph, and I have corrected the first part of that paragraph as follows:

"Modulation of OPN expression offers the possibility to attenuate bone metabolic disorders based on our present understanding of the roles of OPN. In post-menopausal women with diabetes, OPN might be a useful marker for bone fracture and a worsening lipid profile (6). Our results provide new insight into the role of eWAT-secreted OPN in bone homeostasis during obesity, suggesting OPN-targeting strategies for prevention of bone loss in cases of a high-fat diet. ..."

Additional Comment (A). The immunohistochemical staining for OPN is of poor quality, and it is not clear what is actually being stained, no less quantified. It may be an antibody problem - the R&D Systems antibody is most commonly used, not the ABCAM antibody. For example, in Figure 6A, it is difficult to see what is actually stained (cement lines where OPN is abundant, as cited correctly by the authors, could be used as a positive control). Moreover, the authors claim quantification of OPN-positive cells in the histogram data in the panel at the right. What cells? Unless I missed it, I could not find this anywhere in the figure legend or in the text. Given that the quantification apparently comes from identifying these cells in bone and counting them, then I and the readers would surely like to see such a stained cell, and which cell it in fact is. I cannot see the labeled cells, no less count them. Hopefully the marrow megakaryocytes are not being counted, which is likely nonspecific staining.

Additional Comment (B). The authors should address better the field of circulating OPN linked to cancer biology. There are many papers in that field dealing with OPN concentrations in blood, publications germane to the discussions here.

Response to reviewers (NCOMMS-20-33071A)

The reviewers' insightful comments are very helpful for improving our work further towards publication. Based on these comments/suggestions, we have performed an additional experiment and addressed the concerns at a point-by-point basis as below. The authors sincerely hope that our replies have adequately addressed all concerns.

Note: Reviewers' comments are highlighted in blue and shown in italic. (Response Fig./Table) means the figure/table used just to answer the questions in our response letter. (Page, Line, Fig.) means the place(s) where the changes are made in the revised manuscript.

REVIEWER #2

The authors have addressed all questions.

Reply: Many thanks for confirming that our revisions are satisfactory.

REVIEWER #3

The authors have thoroughly addressed my concerns. Two minor comments are recommended to improve the impact of the final manuscript.

Reply: We appreciate reviewer's confirmation with two minor comments for improving our work towards publication. We have added relevant information as specified following.

1. 2-way ANOVA results should be reported throughout to promote accurate interpretation of the data. For example, all panels of Fig.3B should be analyzed by 2-way ANOVA (Diet x eWATx) to determine the p values for the main and interaction effects for diet type and eWATx. 2-way ANOVA P-values for main and interaction effects should be reported in the figure legends or as a supplemental table(s) for relevant graphs. This includes Fig.3B,D; Fig4D,E; Fig.6D; Fig.7I,J,L; etc.

Reply: Thanks for the advice and we have included supplemental tables for the accurate interpretation of data statistics with Two-way ANOVA as shown in the "Supplementary information for Tables".

2. Fig.6A. Please add the units to the graph.

Reply: Added as suggested (Fig. 6A).

Reviewer #4 (Remarks to the Author):

Response to Reviewer #1 rebuttal

Comment 1 (explain the small groups of animals used) - The authors have adequately explained the requested sample size calculations.

Reply: Thanks for your confirmation on our justification for sample size calculation.

Comment 2 (why is there high OPN at 12 weeks) - The authors have explained clearly why large amounts of OPN occur at 12 weeks. This correlates well, as expected, with high macrophage numbers in the crown like structures (CLSs) surrounding the eWAT of the HFD mouse group.

Reply: Thanks for your confirmation and accepting our revision.

Comment 3 (revise Intro to focus more on OPN and bone density) - There are some major issues with the added paragraph, and I have corrected as follows:

"OPN secreted predominantly by osteoblasts and osteocytes is an abundant noncollagenous phosphoprotein of the extracellular matrix of bone. There it serves to regulate matrix mineralization and bone cell (osteocyte and osteoclast) attachment (16-19). OPN is highly concentrated at cement lines where pre-existing and newly formed bone integrate (mineral-matrix interactions), and at bone surfaces in the so-called lamina limitans structure surrounding osteocytes and underlying osteoclasts (cell-matrix interactions) (17). OPN inhibits mineralization by stabilizing mineral precursor phases and/or by binding to crystal surfaces (doi: 10.1016/j.bone.2020.115447), and OPN-deficient mice possess increased mineral content and crystallinity (4,5). OPN-dependent intracellular signaling is seen during initial osteoclast attachment to bone surfaces and during the osteoclastic resorption phase (3), and thus OPN appears to be an important player in the bone remodeling cycle. To date, the systemic effects of OPN on bone homeostasis have not been well studied, especially with regards to the biology of eWAT. Here we further investigate links among bone, OPN, and eWAT."

Reply: Thanks for your help with improvement of above statement and we have revised that paragraph accordingly (Page 4, Line 64 - 77).

Comment 4 (highlight in Discussion potential therapeutic strategies for modulating OPN for bone healing). There are some issues with the added paragraph, and I have corrected the first part of that paragraph as follows:

"Modulation of OPN expression offers the possibility to attenuate bone metabolic disorders based on our present understanding of the roles of OPN. In post-menopausal women with diabetes, OPN might be a useful marker for bone fracture and a worsening lipid profile (6). Our results provide new insight into the role of eWAT-secreted OPN in bone homeostasis during obesity, suggesting OPN-targeting strategies for prevention of bone loss in cases of a high-fat diet. ..."

Reply: Thanks for your help with improving our statement and we have updated that accordingly in our revised version (Page 19, Line 422 - 433).

Additional Comment (A). The immunohistochemical staining for OPN is of poor quality, and it is not clear what is actually being stained, no less quantified. It may be an antibody problem - the R&D Systems antibody is most commonly used, not the ABCAM antibody. For example, in Figure 6A, it is difficult to see what is actually stained (cement lines where OPN is abundant, as cited correctly by the authors, could be used as a positive control). Moreover, the authors claim quantification of OPN-positive cells in the histogram data in the panel at the right. What cells? Unless I missed it, I could

not find this anywhere in the figure legend or in the text. Given that the quantification apparently comes from identifying these cells in bone and counting them, then I and the readers would surely like to see such a stained cell, and which cell it in fact is. I cannot see the labeled cells, no less count them. Hopefully the marrow megakaryocytes are not being counted, which is likely nonspecific staining.

Reply: Thank you very much for the professional comment. We have replenished the immunohistochemical staining for OPN using another antibody and quantified the positive stained area as you suggested (Response Fig. 3). We have also supplemented these data into the revised version (Fig. 6a).

Fig. 1. Representative images of immunohistochemical staining OPN (left) and OPN positive stained area (right, $n = 3$ biologically independent samples) from proximal tibiae of NFD- and HFD-fed groups, Scale bar: 50 μm .

Additional Comment (B). The authors should address better the field of circulating OPN linked to cancer biology. There are many papers in that field dealing with OPN concentrations in blood, publications germane to the discussions here.

Reply: Suggestion is well-taken and we have supplemented relevant information into the Discussion of the revised version as following, including related publications.

“OPN imparting matricellular phosphoglycoprotein functions as an adaptor and modulator for cell matrix interactions, in favoring of cell migration, cell adhesion, and extracellular matrix invasion (1). OPN in some sense ranks germane to the lower overall and disease-free survival in versatile tumors, such as hepatocarcinoma, breast cancer, lung cancer, and prostatic carcinoma, while OPN level is closely correlated with the tumor metastasis, grade, poor prognosis, and early tumor recurrence, owing largely to the soluble OPN allowing paracrine signaling on other tissues (2).” We have supplemented these points into the Discussion section of the revised manuscript (Page 17 - 18, Line 379 - 386).

Reference: (also included in the revised manuscript)

1. G. Musso, E. Paschetta, R. Gambino, M. Cassader, F. Molinaro, Interactions among bone, liver, and adipose tissue predisposing to diabetes and fatty liver.

Trends in molecular medicine **19**, 522-535 (2013).

2. G. F. Weber, G. S. Lett, N. C. Haubein, Osteopontin is a marker for cancer aggressiveness and patient survival. *British journal of cancer* **103**, 861-869 (2010).